# Nanoparticle elasticity affects systemic circulation lifetime by modulating adsorption of apolipoprotein A-I in corona formation

Mingyang Li [1,2,3], Xinyang Jin [1,2,3], Tao Liu [1,2,3], Feng Fan [1,2,3], Feng Gao[1,2,3], Shuang Chai [1,2,3] & Lihua Yang [1,2,3]✉

Nanoparticle elasticity is crucial in nanoparticles' physiological fate, but how this occurs is largely unknown. Using core-shell nanoparticles with a same PEGylated lipid bilayer shell yet cores differing in elasticity (45 kPa – 760 MPa) as models, we isolate the effects of nanoparticle elasticity from those of other physiochemical parameters and, using mouse models, observe a non-monotonic relationship of systemic circulation lifetime *versus* nanoparticle elasticity. Incubating our nanoparticles in mouse plasma provides protein coronas varying non-monotonically in composition depending on nanoparticle elasticity. Particularly, apolipoprotein A-I (ApoA1) is the only protein whose relative abundance in corona strongly correlates with our nanoparticles' blood clearance lifetime. Notably, similar results are observed when above nanoparticles' PEGylated lipid bilayer shell is changed to be non-PEGylated. This work unveils the mechanisms by which nanoparticle elasticity affects nanoparticles' physiological fate and suggests nanoparticle elasticity as a readily tunable parameter in future rational exploiting of protein corona.

[1] Hefei National Research Center for Physical Sciences at the Microscale, University of Science and Technology of China, Hefei 230026 Anhui, China. [2] CAS Key Laboratory of Soft Matter Chemistry, University of Science and Technology of China, Hefei 230026 Anhui, China. [3] School of Chemistry and Materials Science, University of Science and Technology of China, Hefei 230026 Anhui, China. ✉email: lhyang@ustc.edu.cn

Nanoparticles are promising for diverse biomedical applications including vaccines[1,2], diagnostics and imaging[3], and drug delivery[4,5]. To improve nanoparticle's performances in vivo, researchers have focused intensive efforts on tuning nanoparticles' physiochemical parameters including size, shape, and surface chemistry[6]. Nanoparticle elasticity, a previously underexplored physiochemical parameter, has recently been found to play crucial roles in nanoparticles' physiological performances as well. Similar as nanoparticles' other physiochemical parameters like size, shape and surface chemistry[7], nanoparticle elasticity significantly affects nanoparticles' cellular uptake efficiency[8–11], mode of cell internalization[12,13], systemic circulation lifetime[14–17] and biodistribution[8,9,11,13,15–19]. Despite of the accumulating evidences that support nanoparticle elasticity's crucial roles in nanoparticles' physiological fates, the underlying mechanism by which nanoparticle elasticity does so is remain, however, unknown.

Rapidly (within 0.5 min[20]) after a nanoparticle's introduction into a living system, protein corona forms over the nanoparticle's surface through adsorption of proteins from the environmental biofluids[20–22]. Even coating a nanoparticle with polyethylene glycol (PEG), the current gold-standard stealth material for repelling protein adsorption, cannot completely eliminate the formation of protein corona[23–26] but instead selectively suppress the adsorption of certain plasma proteins[23,26]. Of note, the extrinsic physiochemical properties a nanoparticle acquires through protein corona formation, rather than the intrinsic physiochemical properties determined by the nanoparticle's physiochemical parameters (like size, shape, and surface chemistry), determine the nanoparticle's biological identity[27–29], thereby modulating the particle's overall pharmacological and toxicological profiles[20] and potential therapeutic and/or diagnostic functionalities[28,29]. For example, for active targeting nanoparticles, protein corona may mask the targeting ability of their active targeting ligands[24,30–32], serve as intermediary agents for directing them toward their targets[33,34], or exert negligible effects on this aspect[35,36]. To enable future rational exploiting of protein corona, extensive research efforts have been directed towards the relationships between protein corona and nanoparticle physiochemical parameters, which unlike protein corona can be directly and rationally controlled in nanoparticle preparation. To date, nanoparticle physiochemical parameters are known to crucially affect protein corona include size[2,37–39], shape[37,40], and surface chemistry[21,25,28,39] (including surface hydrophobicity[21], charge[39], density of surface-conjugated PEG[25]). Nevertheless, no report to our best knowledge has systematically examined the effects of nanoparticle elasticity on protein corona, despite that nanoparticle elasticity is crucial in nanoparticle's physiological fate (both in vitro[8,9,11–13] and in vivo[8,14–19]) and that protein corona formation on nanoparticles of differing elasticity may lead to different changes in nanoparticle size —protein corona formation on hard nanoparticles is manifested as an increase in particle mean diameter[20,22,33,41] whereas that on liposomes (known to be elastic and soft) can lead to either an increase[26] or reduction[24,26] in liposome mean diameter—and can result in different surface coverages of some protein family groups[42].

In this work, to unveil the mechanisms by which nanoparticle elasticity modulates nanoparticles' physiological performances, we examine whether nanoparticle elasticity affects protein corona and whether there exist certain plasma proteins crucial in nanoparticles' elasticity-dependent fate. Using core-shell nanoparticles with a same PEGylated lipid bilayer shell yet cores differing in elasticity (45 kPa–760 MPa) as models, we isolate the effects of nanoparticle elasticity from those of other physiochemical parameters. Our plots on systemic circulation lifetime in

mouse versus nanoparticle elasticity reveal a non-monotonic relationship, rather than a monotonic one as claimed by prior reports (Supplementary Table 1), though plots of normalized cellular uptake versus nanoparticle elasticity lack a clear trend as did in prior reports (Supplementary Table 2). Incubating our model nanoparticles in mouse plasma results in protein coronas whose compositions varies non-monotonically depending on nanoparticle elasticity. In particular, apolipoprotein A-I (ApoA1) is the corona protein which preferentially accumulates over nanoparticles of intermediate elasticity (75–700 kPa), outcompeting other plasma proteins thereon, and whose relative abundance in corona exhibits strong positive correlation with our nanoparticles' systemic circulation lifetime in mice. Notably, when changing the lipid bilayer shell of above PEGylated nanoparticles to be natural yet non-PEGylated, we observe similar corona patterns for particles of same elasticity, suggesting overwhelming effects of nanoparticle elasticity over surface chemistry. This work demonstrates modulating protein corona as a mechanism by which nanoparticle elasticity affects nanoparticles' systemic circulation lifetime and suggests nanoparticle elasticity as a readily tunable physiochemical parameter for future rational exploiting of protein corona.

## Results and discussion

**Model nanoparticles for isolating the effects of elasticity.** To examine whether nanoparticle elasticity affects nanoparticles' physiological fate and protein corona, we first need model nanoparticles that can enable us to isolate the effects of nanoparticle elasticity from those of other nanoparticle physiochemical parameters (including size, shape and surface chemistry). To this end, we used core-shell nanoparticles that have a same lipid bilayer shell while hydrogel nanoparticle cores differing in crosslinking density as our model nanoparticles (Fig. 1a), considering that a hydrogel particle has its elasticity readily tunable by varying the crosslinking density within the hydrogel and that a lipid bilayer-coated nanoparticle has its surface chemistry controlled only by the lipid composition of its lipid bilayer shell[43]. As a proof-of-concept, we prepared our model nanoparticles by preloading a PEGylated liposome composed of DOPC: DSPE-PEG$_{2000}$ = 90: 10 (mass ratio) (hydrodynamic diameter d ~ 160 nm, zeta-potential ζ ~ −28.7 mV) (Fig. 1e) with a hydrogel-precursor solution containing acrylamide (monomer), N,N'-methylenebis (acrylamide) (crosslinker), and Irgacure 2959 (photo-initiator) (Supplementary Fig. 1) and the exposing the resulting dispersion successively to sodium ascorbate (a scavenger of free radicals) and ultraviolet light irradiation to selectively crosslink the hydrogel-precursor solution within the interior space of the liposome (Fig. 1b). Successful gelation within liposome was confirmed by the inability of Triton X-100 (a detergent good at solubilizing lipid bilayer) to eliminate the Tyndall's effect or the dynamic light scattering signals at sizes of ~100 nm in the nanogel@lipid particle's dispersion (Supplementary Fig. 2). Assuming a nanogel@lipid particle and its corresponding bulk gel prepared with a same hydrogel-precursor solution have the same elasticity, we named the nanogel@lipid particle directly with Young's modulus of the corresponding bulk gel followed by "@lipid" (Fig. 1c). For example, 75 kPa@lipid was used as the name for the hydrogel@lipid nanoparticle in which the hydrogel core was prepared with a hydrogel-precursor solution that offered a bulk hydrogel with a Young's modulus of 75 kPa. By simply adjusting the monomer-to-crosslinker weight ratio in hydrogel-precursor solution, we readily tuned the elasticity of the resulting nanogel@lipid and consequently obtained a series of nanogel@lipid nanoparticles with elasticity ranging from 75 kPa to 1700 kPa (Fig. 1c and Supplementary Fig. 3). Besides, to further

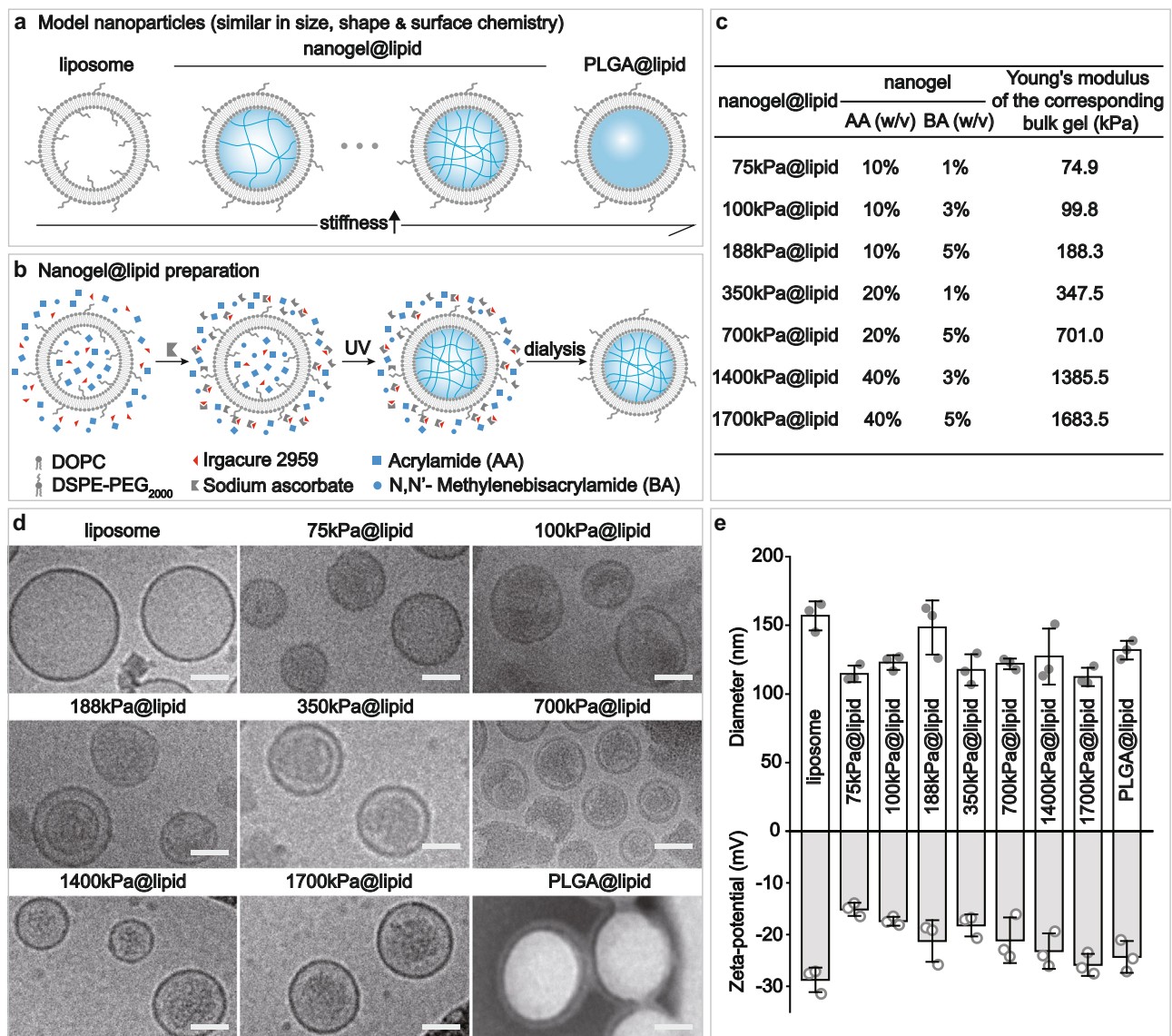

**Fig. 1 Our PEGylated model nanoparticles. a** Schematic illustration on our model nanoparticles, which share a core-shell structure with a lipid bilayer shell of same composition while a core of tunable elasticity. **b** Schematic illustration on the preparation of a nanogel@lipid particle, whose elasticity can be tuned by adjusting its nanogel core's crosslinking density. **c** Summary on the Young's moduli of bulk gels which were prepared with corresponding hydrogel-precursor solutions used for preparing our nanogel@lipid particles. **d** Cryo-electron microscope (Cryo-EM) images of our liposome and nanogel@lipid nanoparticles, and transmission electron microscope (TEM) images of PLGA@lipid negatively stained with uranyl acetate. For each sample, microscopy images were taken in ≥5 different microscopy fields of view, and consistent results were observed. Scale bar = 50 nm. **e** Hydrodynamic diameters and surface zeta-potentials of our model nanoparticles. Bar heights are reported as average ± standard deviation ($n = 3$ independent experiments).

expand the elasticity range of our model nanoparticles both to the extreme soft and stiff ends, we further added to our nanogel@lipid nanoparticles two extra nanoparticles of same surface chemistry but distinct elasticity, which are liposomes (i.e., DOPC: DSPE-PEG$_{2000}$ = 90 : 10) and poly(lactic-co-glycolic acid) (PLGA) nanoparticle coated with a same lipid bilayer (i.e., DOPC: DSPE-PEG$_{2000}$ = 90: 10) (Fig. 1a), as liposomes and poly(lactic-co-glycolic acid) (PLGA) nanoparticles——both are widely used drug carrier platforms[44]——have Young's moduli of 45 kPa[12] (Supplementary Table 4) and 760 MPa[43], respectively. Under Cryo-electron microscopy (Cryo-EM) and transmission electron microscope (TEM) (Fig. 1d), all our model nanoparticles appeared spherical and exhibited a core-shell structure. Moreover, our nanoparticles were similar in size (d, 112 ~ 157 nm) and surface zeta-potential (ζ, −15.1 ~ −28.7 mV) (Fig. 1e). It should be noted that, for our model nanoparticles which are coated with

a same lipid bilayer shell, their surface chemistry (including chemistry composition like PEG density and surface charge) is determined by the lipid composition of their lipid bilayer shell, rather than their readings of zeta-potential (Supplementary Fig. 5). Collectively, these results suggest successful preparation of nanoparticles that are similar in shape, size, and surface chemistry but differ significantly in elasticity (45 kPa–760 MPa). Moreover, for both nanogel@lipid and PLGA@lipid particles, the lipid bilayer coating is very stable, remaining intact even after 7-day incubation in fetal bovine serum (FBS)-supplemented phosphate buffered saline (PBS) (v./v., 50%) (Supplementary Fig. 6).

**Dependence of blood circulation lifetime on nanoparticle elasticity.** In prior reports on how nanoparticle elasticity significantly affects nanoparticles' physiological fate, systemic

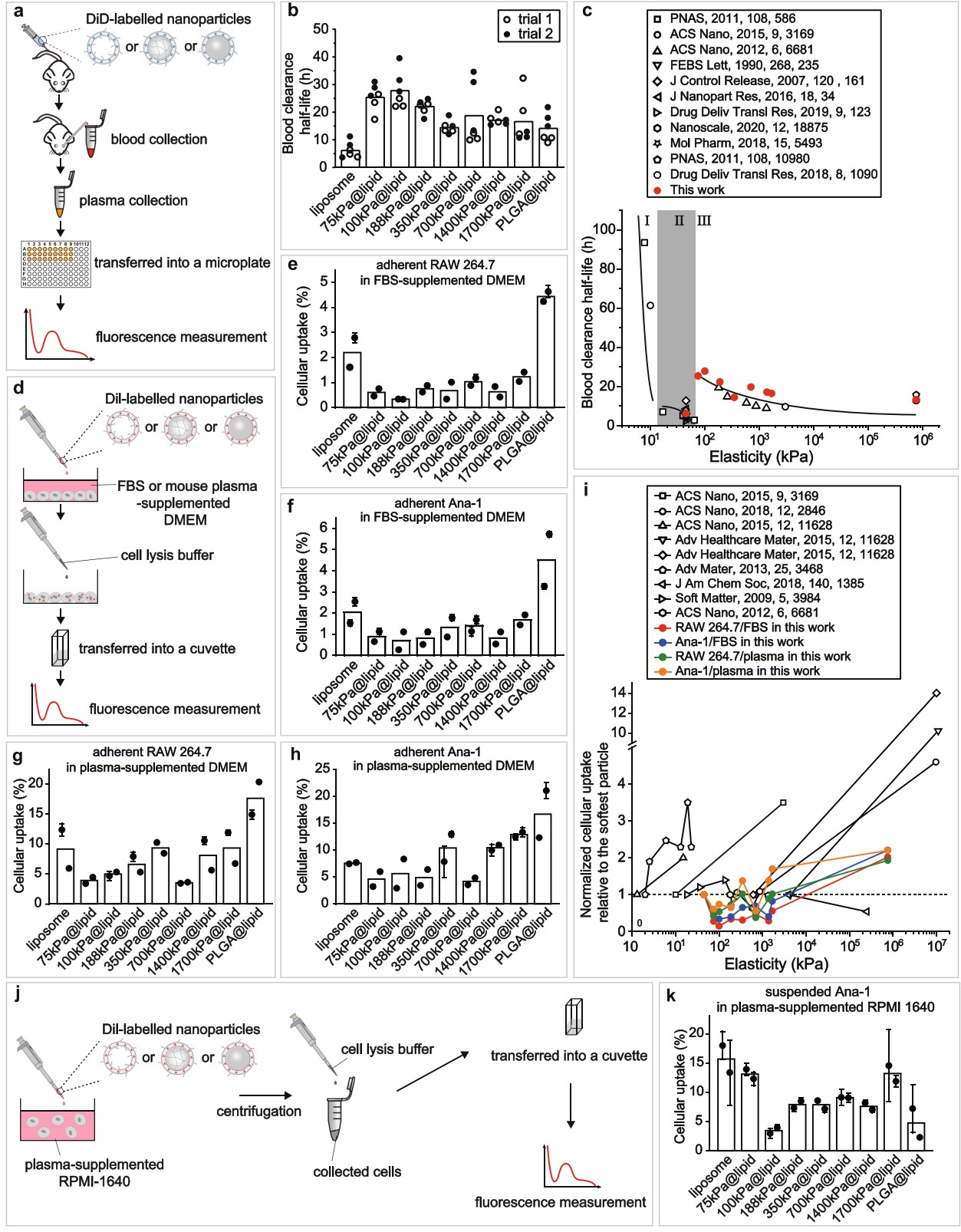

circulation lifetime (Supplementary Table 1) and cellular uptake efficiency (Supplementary Table 2) present the two most intensively studied aspects of physiological fate. Therefore, we herein focused on systemic circulation lifetime and cellular uptake efficiency (Fig. 2), as did prior reports on this topic; still, we have to note that, beside these two above, other aspects of physiological

fate such as biodistribution profile are important as well and deserve future research efforts.

We started by characterizing the systemic circulation lifetimes of our model nanoparticles (Fig. 2a, b). Specifically, we injected our nanoparticles (labelled with DiD, a membrane dye) into ICR mice through tail vein and, at different time-points after injection,

**Fig. 2 Effects of nanoparticle elasticity on blood circulation lifetime and cellular uptake. a** Schematic illustration on blood circulation test for obtaining blood retention profiles for our nanoparticles. **b** Blood clearance half-lives of our model nanoparticles. Bar heights are reported as averages of two independent trials ($n = 3$ biologically independent mice in each independent trial). **c** Plot on the relationship of blood clearance half-life versus nanoparticle elasticity, using results from this work and prior reports on this topic. **d** Schematic illustration on in vitro cellular uptake assays. Uptake efficiency of nanoparticles by murine macrophage **e** RAW264.7 and **f** Ana-1 cells. Bar heights are reported as averages of two independent trials, while data points are reported as average ± standard deviation ($n = 3$ in each independent trial). Uptake efficiency of our nanoparticles by adherent **g** RAW264.7 and **h** Ana-1 cells in mouse plasma-supplemented culture medium. Bar heights are reported as average of two independent trials, while data points are reported as average ± standard deviation ($n = 3$ in each independent trial). **i** Plots on the relationship of normalized cellular uptake relative to the softest particle in a same study versus nanoparticle elasticity, using results from this work and prior reports on this topic. **j** Schematic illustration on in vitro cellular uptake assays by suspended Ana-1 in mouse plasma-supplemented culture medium. **k** Uptake efficiency of our nanoparticles by suspended Ana-1 in mouse plasma-supplemented culture medium. Bar heights are reported as average of two independent trials, while data points are reported as average ± standard deviation ($n = 3$ in each independent trial).

collected blood samples to monitor DiD's content therein (Fig. 2a). Based on the resulting blood retention profiles (Supplementary Fig. 8), we calculated nanoparticles' blood clearance half-lives ($t_{1/2}$) using a two-compartment model (Fig. 2b). Our liposome and PLGA@lipid, which were included to define the boundaries of elasticity range for our model nanoparticles, exhibited the shortest and second shortest clearance half-lives, respectively. Of note, the blood clearance half-lives of liposome (6.2 h) and PLGA@lipid (14.2 h) are consistent with those in prior reports on PEGylated liposomes (ranging from 2.17 h to 12.8 h) (Supplementary Table 8)[45–47] and PLGA nanoparticles (ranging from 10.8 h to 15.8 h)[48,49] and our observed shorter blood circulation lifetime for liposome than PLGA@lipid is supported by prior observations that empty liposomes have shorter lifetimes than lipid bilayer-coated hard nanoparticles (Supplementary Table 9 and Supplementary Fig. 9), both of which confirms the reliability for our systemic circulation assays. Compared with liposome and PLGA@lipid, our nanogel@lipid particles - which fell into the intermediate section of the elasticity range for our model nanoparticles - unanimously exhibited longer clearance half-lives, ranging from 16.5 h for 1700 kPa@lipid to 27.9 h for 100 kPa@-lipid. Clearly, softer nanoparticles do not necessarily exhibit longer systemic circulation, which contrasts significantly with the consistent claim of "longer systemic circulation for softer nanoparticles than their stiffer counterparts" in prior reports (Supplementary Table 1)[14,15,17]. To get a bird's eye view on how nanoparticle elasticity affects nanoparticle systemic circulation lifetime, we compiled into a single plot the relationship of nanoparticle blood clearance half-life versus nanoparticle elasticity not only for our model nanoparticles but also particles in previous reports on this topic[14,15,17] (Fig. 2c). To our surprise, the as-compiled plot surprisingly revealed a non-monotonic relationship that further segmented into three distinct regions depending on nanoparticle elasticity, with an intermediate region located at elasticity of 15–75 kPa. Specifically, these three distinct regions (Fig. 2c) are Region I where nanoparticle elasticity is <15 kPa, Region II where nanoparticle elasticity is 15–75 kPa, and Region III where nanoparticle elasticity is >75 kPa. Within each region, lower particle elasticity does correspond to longer blood clearance lifetimes, as claimed in prior reports[14,15,17]. Nevertheless, across different regions, particles in Region II (e.g., our liposome, very soft hydrogels[15]) consistently exhibited the shortest clearance half-lives (<8 h). Reasons why prior studies on this topic have instead claimed a monotonic relationship between blood circulation lifetime versus particle elasticity (Supplementary Table 1) are: (1) these prior studies compared the performances of their own particles alone, and (2) these prior studies unconsciously used particles from a same Region or compared particles from Region I with those from Region II or III (i.e., none of them compared particles from Region II with those from Region III) (Supplementary Table 10). Obviously, our nanoparticles provided an

elasticity range wide enough to span Region II to Region III and exhibited a non-monotonic yet clear dependence of systemic circulation lifetime on nanoparticle elasticity, thereby offering us a nice model system to study whether nanoparticle elasticity determines nanoparticles' blood circulation lifetime by modulating protein corona.

**Ambiguous effects of nanoparticle elasticity on cellular uptake.** We next characterized our nanoparticles' cellular uptake efficiency (Fig. 2d–k). Adherently cultured cells are good models for simulating resident cells in organs and, in the literature, exposing a nanoparticle to adherently cultured cells is a widely adopted practice for characterizing the particle's cellular uptake efficiency[8,13,17]. Hence, we used adherently cultured murine macrophage RAW 264.7 and Ana-1 cells to simulate resident phagocytes in organs of the reticuloendothelial system (RES)[50,51] (e.g., liver and spleen) that are responsible for clearing blood-borne biological debris and foreign bodies. Specifically, we co-incubated nanoparticles (labelled with DiI, a membrane dye) with adherent cells in fetal bovine serum (FBS)-supplemented DMEM (Dulbecco's Modified Eagle Medium) medium, washed the as-treated cells to remove freely floating nanoparticles, and then lysed the resulting cells to quantify the dye content therein and thereon (Fig. 2d). Our results (Fig. 2e, f) showed that, with both adherent RAW 264.7 and Ana-1 cells in FBS-supplemented DMEM media, PLGA@lipid and liposome exhibited the highest and second highest cellular uptake, respectively, while the nanogel@lipid particles exhibited appreciably lower cellular uptake; nevertheless, there lacked a clear dependence of cellular uptake efficiency on nanoparticle elasticity. Since mouse models are commonly used for characterizing the systemic circulation lifetime for nanoparticles and were used for this purpose for our nanoparticles (Fig. 2a, b), we further replaced FBS with mouse plasma in above cellular uptake assays to better simulate the environmental fluid present when a nanoparticle interacts with a resident phagocyte in vivo but observed a similar lack of clear dependence of cellular uptake efficiency on nanoparticle elasticity (Fig. 2g, h). Using the softest nanoparticle in a study as the reference for other particles in the same study, we normalized the cellular uptake for nanoparticles in different studies and compiled into a single plot the resulting normalized cellular uptake versus nanoparticle elasticity (Fig. 2i). Based on the resulting compiled plot, one may roughly claim higher cellular uptake for stiffer nanoparticles when comparing nanoparticles with elasticity of >$10^6$ kPa to those with elasticity of <$10^6$ kPa; nevertheless, when comparing particles with elasticity of <$10^6$ kPa, it remained difficult to conclude whether stiffer or softer nanoparticles or those of intermediate elasticity have higher cellular uptake, a similarly unfortunate situation as summarized based on prior reports (Supplementary Table 2)[8,9,11,13,16,17,19]. On the other hand, cells

in blood are in suspended state and present the cells a nano-particle inevitably encounters immediately after the particle's intravenous administration. Therefore, we further used sus-pended Ana-1 cells to simulate cells in blood like the circulating phagocytes and carried out similar cellular uptake assays (Fig. 2j). With suspended Ana-1 cells in plasma-suspended RPMI 1640 media, the cellular uptake efficiencies of our nanoparticles varied appreciably and non-monotonically with increasing nanoparticle elasticity (Fig. 2k), with our liposome and 100 kPa@lipid exhi-biting the highest and lowest cellular uptake efficiencies, respec-tively. Unfortunately, as previous reports normally use adherent cells to probe the effect of nanoparticle elasticity on cellular uptake, we could not compare these results on our nanoparticles' uptake by suspended cells with that in previous reports, neither could we claim there exists or lacks a clear dependence of uptake by suspended cells on nanoparticle elasticity. Due to the lack of a clear dependence on nanoparticle elasticity for nanoparticles' uptake by adherent cells (Fig. 2i) and the uncertainty in whether a clear dependence on nanoparticle elasticity ever exists for nano-particles' uptake by suspended cells (Fig. 2k), we in the following parts of this work neglected aforementioned questions on how nanoparticle elasticity affects cellular uptake but instead directed our focus toward unveiling the mechanisms underlying the clear yet non-monotonic relationship between blood circulation life-time versus nanoparticle elasticity.

**ApoA1 crucial in elasticity-dependence of blood circulation lifetime**. To study how nanoparticle elasticity affects protein corona, we incubated our nanoparticles in mouse plasma (for 12 h) and collected the resulting particle-protein complexes for subsequent determinations on amount and composition of adsorbed proteins. The total amount of adsorbed proteins on a nanoparticle was quantified with a Bradford Protein Assay Kit, and our results (Fig. 3a) revealed appreciably more adsorbed proteins for nanoparticles on both ends of the examined elasticity range (i.e., liposome and 75 kPa@lipid at the soft end while PLGA@lipid on the stiff end) than those in the middle, with the most and least adsorbed protein amounts observed with PLGA@lipid and 1400 kPa@lipid, respectively, suggesting an elasticity in the intermediate elasticity range of 700–1700 kPa to be optimal for repelling protein adsorption. Composition of protein corona was examined qualitatively using SDS-PAGE (sodium dodecyl sulfate–polyacrylamide gel electrophoresis) and quantitatively with LC-MS (liquid chromatography–mass spec-trometry) (Fig. 3 and Supplementary Figs. 12 and 13). SDS-PAGE analysis revealed that the adsorbed proteins differed apparently in molecular weight depending on nanoparticle elasticity (Fig. 3b). LC-MS analysis revealed that, for all our nanoparticles, surface adsorption preferred smaller plasma proteins (<60 kDa) when sorted by molecular weight (Fig. 3c), negatively charged proteins (pI < 7) when sorted by isoelectric point (pI) (Fig. 3d), and lipoproteins and complement proteins when sorted by biological functions (Fig. 3e). Our nanoparticles' preference for smaller plasma proteins is consistent with prior reports[26,52], and their preference for negatively charged proteins (pI < 7) (Fig. 3d) despite of their negative surface charges is consistent with a prior study on negatively charged silica nanoparticles[38] but contrasts significantly with some recent reports[53–56]. Of note, no matter how adsorbed proteins were sorted, protein corona compositions varied appreciably yet non-monotonically depending on nano-particle elasticity (Fig. 3c–e). Collectively, these results suggest that nanoparticle elasticity affects protein corona both in total amount and in corona protein pattern.

To examine whether there exist specific corona proteins crucial in the relationship of systemic circulation lifetime versus

nanoparticle elasticity, we next identified the proteins most abundant on our model nanoparticles; to confirm the repeat-ability of the results, the proteomics analysis was carried out through two independent trials (Fig. 3f and Supplementary Fig. 13a). Notably, among corona proteins that exhibited a relative abundance of >5% on any of our nanoparticles (i.e., the major corona proteins) (Fig. 3g), ApoA1 is the only one that out-competed other plasma proteins on ≥5 members of our model nanoparticle family (Fig. 3g, Supplementary Figs. 13a, 14), dominating the coronas of nanoparticles with intermediate elasticity of 75–700 kPa (Fig. 3g, Supplementary Figs. 13a and 14). In fact, the observed dominance of ApoA1 in corona for some of our model nanoparticles is surprising at first glance but still understandable since ApoA1 has been frequently discovered as a corona protein with top relative abundance (although the related prior reports are on either liposomes or stiff nanoparti-cles) (Supplementary Fig. 14). Next, we evaluated whether a protein's relative abundance in corona correlates with blood clearance lifetime of nanoparticle, by calculating Pearson's correlation coefficient r between the relative abundance of a protein in corona (Fig. 3g) and blood clearance half-life (Fig. 2b) for our model nanoparticles. Our calculations show that, among the detected major corona proteins (Fig. 3g), ApoA1 was the only one whose relative abundance in corona exhibited strong correlation with blood clearance lifetime of nanoparticle (Fig. 3h), suggesting ApoA1 as a key corona protein in the relationship of systemic circulation lifetime versus nanoparticle elasticity.

Calculations on Pearson's r between the relative abundance of a protein in corona (Fig. 3g) and blood clearance half-lives (Fig. 2b) for our model nanoparticles revealed a strong positive correlation (Pearson's r > 0.6) for ApoA1's relative abundance in corona with nanoparticle blood clearance lifetime (Fig. 3h), suggesting ApoA1 as a dysopsonin. To experimentally check whether ApoA1 is a dysopsonin as indicated by our calculations above, we further examined whether screening/shielding the adsorbed ApoA1 in corona with ApoA1 antibody leads to shortened blood circulation lifetime for nanoparticles over which ApoA1 enjoys high relative abundance in corona, assuming that surface adsorbed ApoA1 molecules are exposed to the environment and not denatured. To this end, we carried out similar blood circulation assays as those performed for obtaining the plot on the relationship between nanoparticle blood circulation lifetime versus nanoparticle elasticity (Fig. 2c) but added an extra step - which is the pre-treatment (for 1 h) with ApoA1 antibody (at a molar ratio of ApoA1 antibody to estimated ApoA1 amount in corona ~1:1) - after the 12-h incubation of a particle with mouse plasma while before the intravenous injection of the particle into healthy mouse for blood retention tests (Fig. 3i). ApoA1 abundance in corona was observed to be ~0% for liposome, ~8% for PLGA@lipid, and ~25% for 188 kPa@lipid (Fig. 3f and Supplementary Fig. 13a). Therefore, in these additional blood circulation assays, liposome (DOPC: DSPE-PEG$_{2000}$ = 90 : 10), 188 kPa@lipid, and PLGA@-lipid were used as the representatives for nanoparticles with quite low ApoA1 abundance in corona, nanoparticles with very high ApoA1 abundance in corona, and nanoparticles with intermedi-ate ApoA1 abundance in corona, respectively. Our addition blood circulation assays (Fig. 3j) show that, for liposome (DOPC: DSPE-PEG$_{2000}$ = 90: 10), pre-treating the liposome-protein complex with ApoA1 antibody barely affected the complex's blood clearance half-life (9.1 h and 9.9 h with and without the ApoA1 antibody treatment, respectively), which is easy to understand considering the quite low abundance (~0%) of ApoA1 in its corona (Fig. 3f and Supplementary Fig. 13a). In stark contrast, for 188 kPa@lipid nanoparticle, pre-treating the nanoparticle-protein complex with ApoA1 antibody significantly shortened the complex's blood clearance half-life (Fig. 3j), from

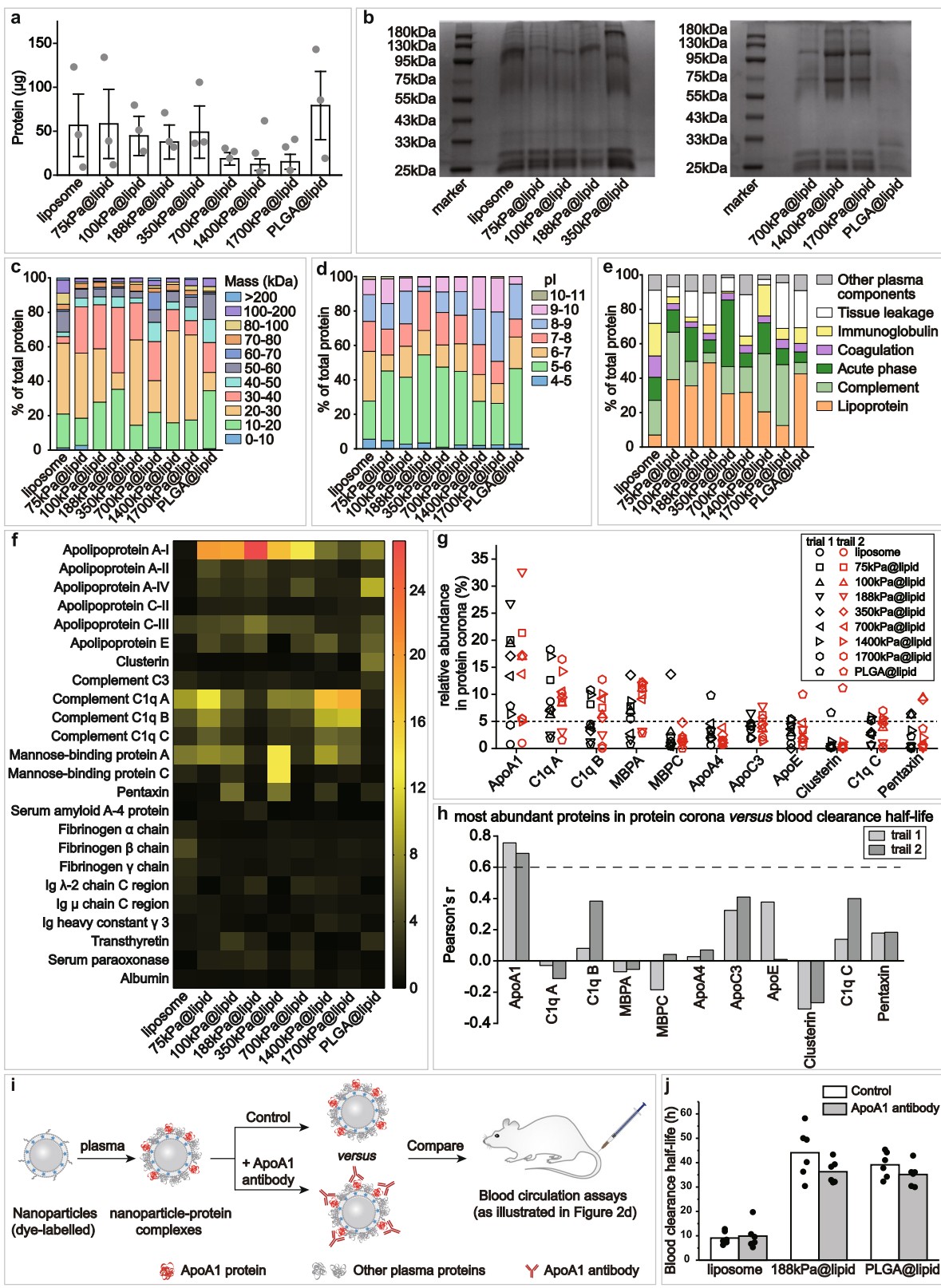

44 h to 36 h in average blood clearance half-life (relatively by ~18%) due to the shielding of adsorbed ApoA1 by ApoA1 antibody. For PLGA@lipid, the pre-treatment with ApoA1 antibody appreciably shortened the complex's blood clearance half-life (from 39 h to 35 h in average blood clearance half-life) (Fig. 3j) but to a less extent than the case with 188 kPa@lipid (relative reduction of ~10% versus ~18%). Interestingly, for these

three model nanoparticles, their order of relative change in blood clearance half-life (0% for liposome, 10% for PLGA@lipid, and ~18% for 188 kPa@lipid) nicely mirrors their ranks of ApoA1 abundance in corona (~0% for liposome, ~8% for PLGA@lipid, and ~25% for 188 kPa@lipid) (Fig. 3f). Clearly, shielding/screening the adsorbed ApoA1 in corona over a nanoparticle shortens the particle's blood circulation lifetime and the relative extent of

**Fig. 3 Effects of nanoparticle elasticity on protein corona. a** Amounts of absorbed proteins on our nanoparticles (kept constant at 1 mg in lipid dose). Data points are reported as average ± standard deviation ($n = 3$ independent experiments). **b** Photographs of SDS-PAGE gels with absorbed proteins on our nanoparticles. The SDS-PAGE gel electrophoresis was only once. Classification of corona proteins according to **c** molecular weight, **d** calculated isoelectric point (pI), and **e** physiological functions. **f** Heat map of the most abundant proteins in protein coronas of our nanoparticles. **g** Distribution of the relative abundance in corona on different particles and **h** Pearson's r between the relative abundance in corona and nanoparticle blood clearance half-life for proteins which exhibited a relative abundance of >5% on at least one nanoparticle. Pearson's r of >0.6 and <−0.6 indicate strong positive and negative correlations, respectively. **i** Schematic illustration on blood circulation tests for examining the effects of pre-screening/pre-shielding the adsorbed ApoA1 in corona with ApoA1 antibody. **j** Blood clearance half-lives of 188 kPa@lipid with and without the adsorbed ApoA1 in corona pre-screened/pre-shielded by ApoA1 antibody, and those of liposome and PLGA@lipid in similar assays are included for reference. Bar heights are reported as average ($n = 6$ biologically independent mice).

---

such shortening is positively related with the particle's relative abundance of ApoA1 in corona.

**Roles of ApoA1 in elasticity-dependent fate of nanoparticles.** As a protein in corona, ApoA1 has been proposed to be able to suppress cellular uptake[57,58], extend systemic circulation lifetime[57], and relieve nanoparticle cytotoxicity and proinflammatory effect[58]; nevertheless, these roles of ApoA1 were observed with either nanoparticles as stiff as solid SiO₂ nanospheres or PEGylated graphene oxide sheets which are 2-dimensional materials (Supplementary Table 11). What roles ApoA1 plays in the relationship of systemic circulation lifetime versus nanoparticle elasticity remains unknown. To address this, we firstly evaluated the interactions of ApoA1 with nanoparticles of differing elasticity with isothermal titration calorimetry (ITC), a powerful technique commonly used for examining the non-specific binding of proteins on nanoparticles[21,59]. Therefore, to understand why ApoA1 was the most abundant corona protein only for nanoparticles of intermediate elasticity (75–700 kPa), we carried out ITC analysis using liposome, 188 kPa@lipid, and PLGA@lipid as the representatives for soft, intermediate elasticity, and stiff nanoparticles, respectively. Upon titration of ApoA1 (Fig. 4a), appreciable thermal effect (specifically, exothermic) was observed but only with 188 kPa@lipid, indicative of strong non-specific binding of ApoA1 on 188 kPa@lipid, but not on our liposome or PLGA@lipid. In stark contrast, in similar ITC assays, titration of bovine serum albumin (BSA) (Fig. 4b), a protein widely used as a representative for dysopsonins[60], revealed negligible thermal effects with all three examined nanoparticles irrespective of their difference in nanoparticle elasticity. Clearly, even in the absence of competitors (e.g., diverse proteins in blood), ApoA1, but not BSA, preferentially binds on nanoparticles of intermediate elasticity as exemplified by 188 kPa@lipid, which may explain why ApoA1 out-competed other plasma proteins in adsorption on nanoparticles with intermediate elasticity of 75–700 kPa (Fig. 3f and Supplementary Fig. 13a).

ApoA1 was the only major corona protein whose relative abundance in corona strongly correlated with our nanoparticles' blood clearance lifetimes (Fig. 3h). To understand the contribution of ApoA1 adsorption to these nanoparticles' systemic circulation, we evaluated the effects of ApoA1 supplementation on the cellular uptake efficiency of our nanoparticles by macrophages (Fig. 4c, d), as blood clearance due to internalization by phagocytes such as macrophages[7] is an important process that, together with many others like the particle size-based clearance via filtration in major organs (Supplementary Table 12), defines the blood clearance lifetime of a nanoparticle. Again, liposome, 188 kPa@lipid, and PLGA@lipid were used as the representatives for soft, intermediate elasticity, and stiff nanoparticles, respectively. Note that phagocytes in blood present a major barrier which a nanoparticle may encounter in vivo[7] and are naturally in suspended state. We hence used murine macrophage Ana-1 cells cultured in suspended state

as the in vitro model for phagocytes in blood and found that, in serum-free media, ApoA1 supplementation resulted in >30% reduction in cellular uptake, as compared to the <5% reduction in cellular uptake by BSA supplementation (Fig. 4c), suggesting ApoA1 as a more effective dysopsonin than BSA in the case of suspended cells. In addition to circulating phagocytes in blood, resident phagocytes in major organs (e.g., Kupffer cells in liver) present another barrier which a nanoparticle may encounter in vivo[7,61]. Unlike circulating phagocytes in blood, resident phagocytes in major organs are naturally in an adherent state. Therefore, we carried out similar cellular uptake assays but using adherent Ana-1 cells as the in vitro models for resident phagocytes in major organs (Fig. 4d) and found that, in serum-free media, ApoA1 supplementation resulted in 43–76% reduction in cellular uptake, as compared to the 2–25% reduction in cellular uptake by BSA supplementation (Fig. 4d), suggesting ApoA1 as a more effective dysopsonin than BSA in the case of adherent cells. Collectively, these observations suggest that, even in the absence of any competitors for adsorption onto nanoparticle surface, ApoA1 is more efficient in suppressing cellular uptake than BSA no matter whether the cells which nanoparticles interact with are suspended or adherent.

Nevertheless, in the interactions of a nanoparticle with cells in vivo, competitors are naturally present not only as proteins in the environmental body fluids but also as corona proteins pre-adsorbed prior to the particle's encountering with the cells. To better simulate the roles of surface adsorbed ApoA1 in the interactions of nanoparticles with cells in vivo, we further examined how screening/shielding ApoA1 in corona with ApoA1 antibody affects the cellular uptake of nanoparticles (Fig. 4e), assuming that surface adsorbed ApoA1 molecules are exposed to the environment and not denatured. To this end, we carried out similar cellular uptake assays as those performed for obtaining the plot on the relationship between nanoparticle cellular uptake efficiency versus nanoparticle elasticity (Fig. 2i) but added an extra step - pre-treating (for 1 h) the plasma protein-nanoparticle complex with ApoA1 antibody (at a molar ratio of ApoA1 antibody to estimated ApoA1 amount in corona ~1:1) - after the 12-h incubation of a particle with mouse plasma while before the exposure of the resulting protein-particle complex to cultured macrophage cells (Fig. 4e). ApoA1 abundance in corona was observed to be ~0% for liposome, ~8% for PLGA@lipid, and ~25% for 188 kPa@lipid (Fig. 3f and Supplementary Fig. 13a). Therefore, in these additional cellular uptake assays, liposome (DOPC: DSPE-PEG₂₀₀₀ = 90 : 10), 188 kPa@lipid, and PLGA@lipid were used as the representatives for nanoparticles with quite low ApoA1 abundance in corona, nanoparticles with very high ApoA1 abundance in corona, and nanoparticles with intermediate ApoA1 abundance in corona, respectively. Our addition cellular uptake assays (Fig. 4f) show that, for liposome (DOPC: DSPE-PEG₂₀₀₀ = 90 : 10), pre-treating the liposome-protein complex with ApoA1 antibody significantly enhanced the complex's uptake by adherent Ana-1 macrophage cells (from

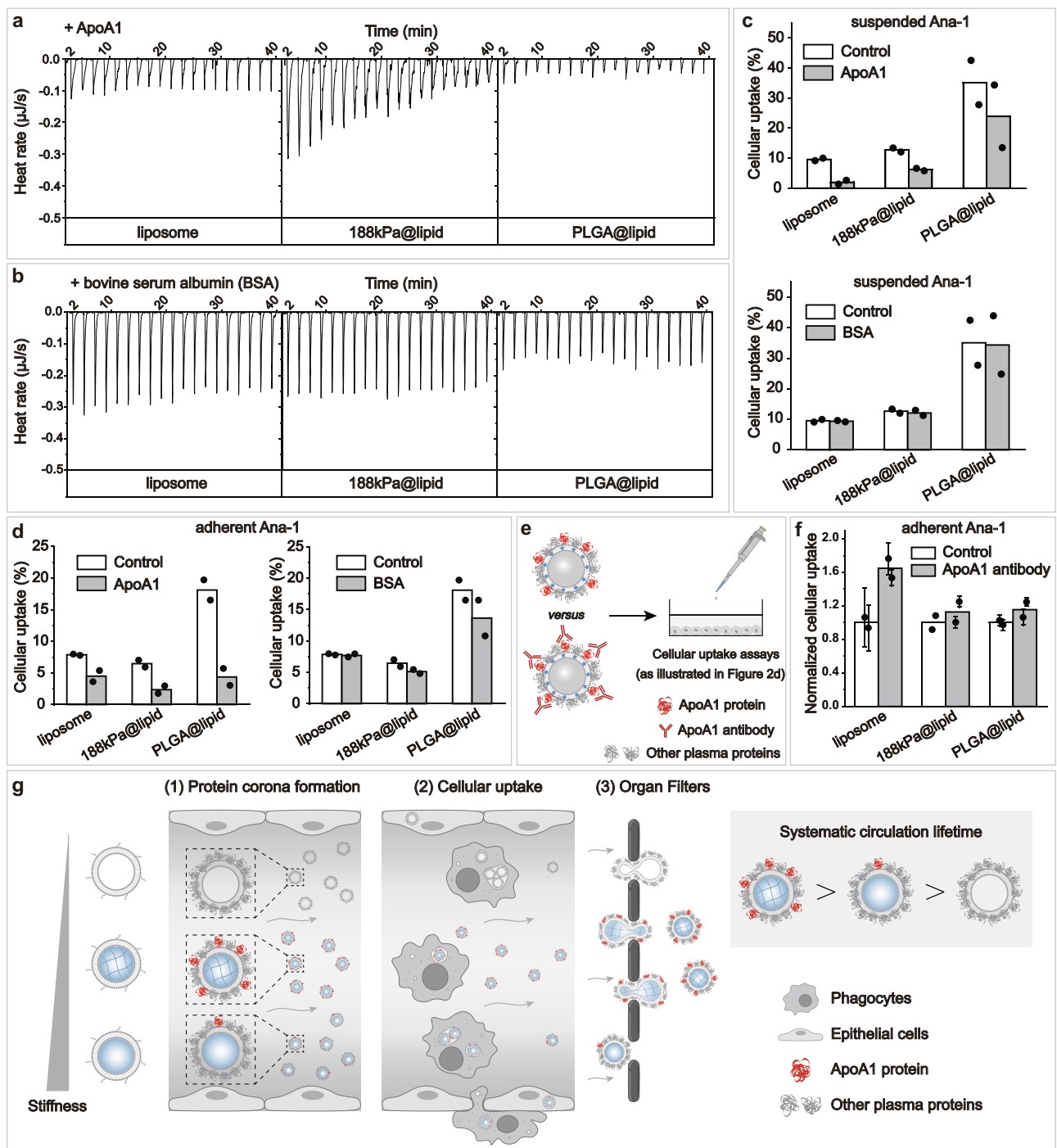

**Fig. 4 Roles of Apolipoprotein A-I (ApoA1) in nanoparticle elasticity-dependence of systemic circulation lifetime.** Raw data from isothermal titration calorimetry (ITC) assays, in which **a** ApoA1 and **b** bovine serum albumin (BSA) solution was titrated into nanoparticle dispersions, using liposome, 188 kPa@lipid, and PLGA@lipid as representatives for nanoparticles with distinct elasticity. **c** Effects of the presence of ApoA1 and BSA (both at 50 μg/mL) on the uptake efficiency of our representative nanoparticles by suspended Ana-1 cells in RPMI-1640. Bar height are reported as average of two independent trials. **d** Effects of the presence of ApoA1 and BSA (both at 50 μg/mL) on the uptake efficiency of our representative nanoparticles by adherent Ana-1 cells in DMEM. Bar height are reported as average of two independent trials. **e** Schematic illustration on cellular uptake assays for examining the effects pre-screening/pre-shielding the adsorbed ApoA1 in corona with ApoA1 antibody. **f** Cellular uptake of a nanoparticle with and without ApoA1 in corona being pre-screened/pre-shielded with ApoA1 antibody, normalized relative to that of the particle without ApoA1 in corona being pre-screened/pre-shielded with ApoA1 antibody. Bar heights are reported as averages of two independent trials, while data points are reported as average ± standard deviation (*n* = 3 in each independent trial). **g** Schematic illustration on (1) ApoA1's preferentiality in adsorbing onto nanoparticles of intermediate elasticity and consequently (2) suppressing their cellular uptake, which together with (3) softer nanoparticles' higher tendency to pass organ filters lead to (right side) longer systemic circulation lifetime for nanoparticles of intermediate elasticity.

9.3 to and 15% due to ApoA1 antibody treatment) (Fig. 4f), likely due to the adsorption of ApoA1 antibody onto our liposome, suggesting ApoA1 antibody as an opsonin capable of effectively promoting a nanoparticle's cellular uptake. For 188 kPa@lipid, pre-treating the nanoparticle-protein complex with ApoA1 antibody appreciably enhanced the complex's uptake by adherent Ana-1 cells but to a much less extent (from 4.1% to 4.7%, relatively by 17% due to ApoA1 antibody shielding) (Fig. 4f), and similar results were observed for PLGA@lipid (from 18% to 21%, relatively by 16.7% due to ApoA1 antibody shielding) (Fig. 4f). The observation that 188 kPa@lipid and PLGA@lipid exhibited less relative enhancement in cellular uptake after ApoA1 antibody pre-treatment than the liposome is attributable to their appreciably higher relative abundances of ApoA1 in corona, whose dysopsonin effects counteracted at least partially with the opsonin effects of ApoA1 antibody. Clearly, shielding/screening the adsorbed ApoA1 in corona enhances cellular uptake of nanoparticles and the relative extent of such enhancement is negatively related with the particle's relative abundance of ApoA1 in corona.

Combined, above results suggest three roles for ApoA1 in the relationship of systemic circulation lifetime versus nanoparticle elasticity (Fig. 4g): (1) ApoA1 preferentially binds with nanoparticles of intermediate elasticity within 75–700 kPa (e.g., 188 kPa@lipid) over those either too soft or too stiff, (2) ApoA1 in corona effectively suppresses uptake of nanoparticles by both circulating and resident cells, and (3) ApoA1 in corona effectively prolongs the systemic circulation lifetime of nanoparticles. These three roles of ApoA1, combined with soft particles' ability to pass organ filters via deformation, may explain why nanoparticles of intermediate elasticity exhibit longer systemic circulation lifetime than their softer and stiffer counterparts (Fig. 2c). Still, we need to note that our model nanoparticle family lacked members with elasticity of Region I (Fig. 2c) - actually synthetic nanoparticles of that soft elasticity as of Region I are very rarely reported in the literature- and we may have therefore missed the mechanisms by which quite soft nanoparticles from Region I acquire their quite long systemic circulation lifetime.

**Our elasticity-corona pattern resisted surface chemistry change.** To exclude the possibility that the observed effects of nanoparticle elasticity on protein corona is exclusive to our PEGylated lipid bilayer-coated nanoparticles, we changed their surface coating from PEGylated to PEG-absent lecithin (lec) bilayer (Fig. 5a–d) but, after incubating the resulting particles (namely, lec liposome, nanogel@lec, and PLGA@lec) in mouse plasma, observed a similar relationship between nanoparticle elasticity and protein corona composition (Fig. 5e–i, Supplementary Fig. 19). Notably, ApoA1 was the only protein that exhibited a relative abundance of >20% on lecithin-coated particle and exhibited a peak content on 188 kPa@lec (Fig. 5i and Supplementary Fig. 19d), as was the case with PEGylated nanoparticles. Such a similarity suggests resistance of our results to change in nanoparticle surface chemistry.

We had also tried to examine whether our results are resistant to change in nanoparticle size but failed in preparing soft nanogel@lipid particles of larger sizes; we extruded the PEGylated liposome preloaded with hydrogel-precursor solutions through nucleopore membrane with larger pore size (400 nm) but found that the resulting nanogel@lipid nanoparticles still exhibited average size of <200 nm (Supplementary Fig. 20), likely due to the higher viscosity of hydrogel-precursor solutions and thus weaker shear force in hydration process than ultrapure water.

In summary, we isolated the effects of nanoparticle elasticity from those of other of physiochemical parameters (i.e., size,

shape, surface chemistry) by preparing core-shell nanoparticles that have a same shell but differ only in core elasticity (45 kPa–760 MPa) and examined the mechanisms by which nanoparticle elasticity modulates the physiological fate of nanoparticles. Special focus was given to systemic circulation lifetime, thanks to the observation of a clear relationship between systemic circulation lifetime versus nanoparticle elasticity; cellular uptake, though extensively investigated in prior reports on the effects of nanoparticle elasticity, was neglected here, due to the lack of a clear relationship between uptake by adherent cells versus nanoparticle elasticity and the uncertainty whether a clear relationship ever exists between uptake by suspended cells versus nanoparticle elasticity. Proteomics analysis revealed ApoA1 as the only plasma protein whose relative abundance in corona ranked the highest for at least 3 members of our model nanoparticle family, out-competing other plasma proteins for nanoparticles of inter-mediate elasticity within 75–700 kPa. Calculations on correla-tion coefficient showed ApoA1 as the only major corona protein whose relative abundance in corona strongly correlates with the blood clearance half-life for our model nanoparticles, suggesting crucial roles of ApoA1 in the relationship between systemic circulation lifetime versus nanoparticle elasticity. Additional results suggest that, in the relationship of systemic circulation lifetime versus nanoparticle elasticity, ApoA1 may function by preferential binding with nanoparticles of intermediate elasticity within 75–700 kPa (e.g., 188 kPa@lipid) over those either too soft or too stiff, by effectively suppressing uptake of nanopar-ticles by both circulating and resident cells, and by efficiently prolonging the systemic circulation lifetime of nanoparticles. These three roles of ApoA1, combined with soft particles' ability to pass organ filters via deformation, may explain why nanoparticles of intermediate elasticity exhibit longer systemic circulation lifetime than their softer and stiffer counterparts (Fig. 2c). Notably, the observed pattern of nanoparticle elasticity versus corona protein was retained when our nanoparticles' surface was changed from PEGylated to non-PEGylated, suggesting overwhelming effects of nanoparticle elasticity over nanoparticle surface chemistry. Still, it is a pity that our model nanoparticle family lacked a member with elasticity of Region I - actually that soft nanoparticles as from Region I are very rarely prepared in the literature- and we may have therefore missed the mechanisms by which quite soft nanoparticles from Region I acquire their quite long systemic circulation lifetime. This work unveiled the roles of ApoA1 in the mechanisms by which nanoparticle elasticity affects nanoparticles' systemic circulation lifetime and suggests nanoparticle elasticity as a readily tunable parameter in future rational exploiting of protein corona.

## Methods

**Materials**. 1,2-dioleoyl-sn-glycero-3-phosphoethanolamine (DOPC) and 1,2-dis-tearoyl-sn-glycero-3-phosphoethanolamine-N-[methoxy(polyethylene glycol) −2000] (DSPE-PEG$_{2000}$) were purchased from Avanti Polar Lipids (Alabama, USA) and used as received without further purification. Lecithin, bovine serum albumin and acrylamide were purchased from Macklin Biochemical (Shanghai, China). Apolipoprotein A1 (ApoA1) was purchased from Sigma-Aldrich (USA), and ApoA1 antibody was purchased from Bioss (Beijing, China) (Catalog number: bs-4573R, Lot number: AD073151). N,N'-methylenebis (acrylamide) and Irgacure 2959 were purchased from aladdin (China). Poly(lactic-co-glycolic) acid (PLGA) (molar ratio of lactic acid to glycolic acid = 75: 25, Mn ~ 40,000 Da) was purchased from Polymtek Biomaterial (Shenzhen, China). DiD (1,1'-Dioctadecyl-3,3,3',3'-Tetramethylindodicarbocyanine Perchlorate) and DiI (1,1'-Dioctadecyl-3,3,3',3'-Tetramethylindocarbocyanine Perchlorate) were purchased from Invitrogen Life Technologies (America). The RAW 264.7 macrophage and Ana-1 macrophage were purchased from Cell Bank of the Chinese Academy of Sciences (Shanghai, China). All reagents were used as received without further purification unless specified otherwise. All animal experiments were conducted in compliance with the guidelines for the care and use of research animals established by the Animal Care and Use Committee at the University of Science and Technology of China.

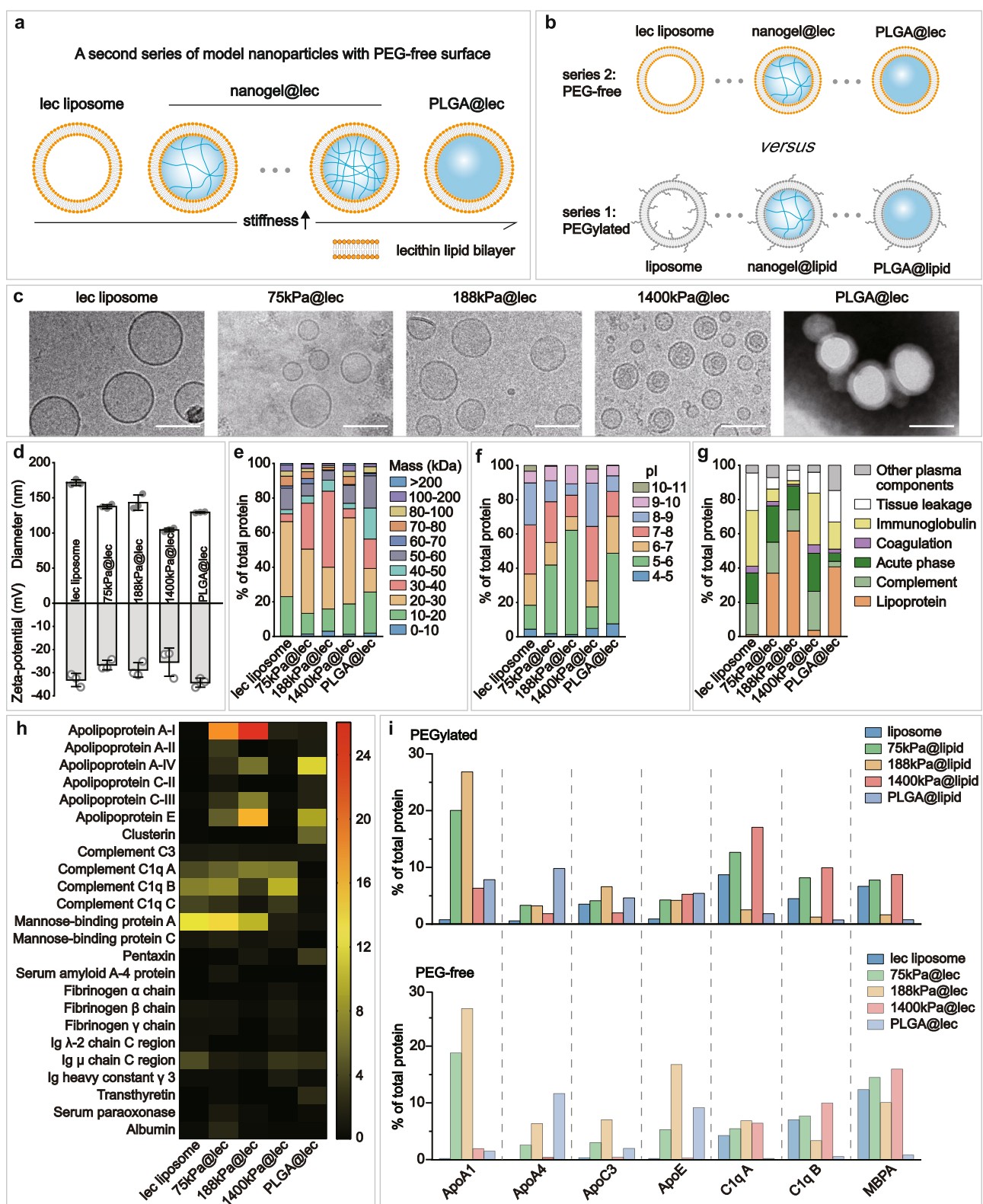

**Preparation of liposome**. DOPC stock solution (20 mg/mL, in chloroform) was mixed with DSPE-PEG₂₀₀₀ stock solution (20 mg/mL, in chloroform) at a mass ratio of DOPC: DSPE-PEG₂₀₀₀ = 90 : 10. The resulting mixtures were dried to a thin film under gentle stream of N₂, and desiccated under vacuum overnight. The resulting thin film was rehydrated with Millipore water at 40 °C for 2 h to a final lipid concentration of 10 mg/mL. The resulting lipid dispersion were subjected to seven freeze−thaw cycles and then extruded through nucleopore membrane with pore size of 0.2 μm (Whatman) for 13 times using a mini-extruder (Avanti Polar

Lipids), which yielded the expected liposome stock dispersion. All liposome stock dispersions were stored at 4 °C prior to use.

**Preparations of the nanogel@lipid particles**. DOPC stock solution (20 mg/mL, in chloroform) was mixed with DSPE-PEG₂₀₀₀ stock solution (20 mg/mL, in chloroform) at a mass ratio of DOPC: DSPE-PEG₂₀₀₀ = 90 : 10. The resulting mixtures were dried to a thin film under gentle stream of N₂, and desiccated under

**Fig. 5 Nanoparticle elasticity overwhelms surface chemistry (PEG presence versus absence) in affecting protein corona.** Schematic illustrations on **a** our PEG-free model nanoparticles, which have a core-shell structure with a lipid bilayer shell composed of lecithin (lec) and a core of tunable elasticity, and **b** a comparison with PEGylated nanoparticles. **c** Cryo-EM images of our lec liposome and nanogel@lec particles and TEM image of our PLGA@lec. For each sample, microscopy images were taken in ≥5 different microscopy fields of view, and consistent results were observed. Scale bar = 100 nm. **d** Hydrodynamic diameter and surface and zeta-potentials of our PEG-free nanoparticles. Bar heights are reported as average ± standard deviation ($n = 3$ independent experiments). Classifications of corona proteins according to **e** molecular weight, **f** calculated isoelectric point (pI), and **g** physiological functions for our PEG-free nanoparticles. **h** Heat map of the most abundant proteins in the coronas on our PEG-free nanoparticles. **i** Comparison on the distribution of the relative abundance in corona on our (top) PEGylated and (bottom) PEG-free nanoparticles for proteins which exhibited a relative abundance in corona of >5% on at least one nanoparticle.

vacuum overnight. The resulting thin film was rehydrated at 40 °C for 2 h to a final lipid concentration of 10 mg/mL with a hydrogel-precursor solution that contained acrylamide (AA) (the monomer), N,N'-methylenebis (acrylamide) (BA) (the cross-linker), and Irgacure2959 (the photo-initiator) in Millipore water (Supplementary Table 3). The resulting lipid dispersion were subjected to seven freeze−thaw cycles and then extruded through nucleopore membrane with pore size of 0.2 μm (Whatman) for 13 times using a mini-extruder (Avanti Polar Lipids), which yielded a liposome dispersion in which both the intra- and extra-vesicular solutions were the hydrogel-precursor mixture. Into the resulting liposome dispersion was added sodium ascorbate (a scavenger of free radicals) (at a molar ratio of sodium ascorbate: Irgacure2959 = 200 : 1) to prevent the extravesicular solution from polymerization. The resultant dispersion was subsequently subjected to irradiation with an ultraviolet (UV) laser (365 nm, for 40 min) and then dialysis against Millipore water to remove free hydrogel precursors and sodium ascorbate, which yielded the expected nanogel@lipid particle stock dispersion. All nanogel@lipid stock dispersions were stored at 4 °C prior to use.

**Preparation of the PLGA@lipid nanoparticle.** PLGA nanoparticle was prepared by a nanoprecipitation process[62]. Briefly, PLGA (75:25, Mn ~40,000) was dissolved into acetone (to a final concentration of 3 mg/mL), and the resulting solution (1 mL) was added dropwise into sterile Millipore water (3 mL), followed by stirring in open air for 10 h to evaporate the acetone, which yielded the expected PLGA nanoparticle.

The as-prepared PLGA nanoparticle was then coated with a lipid bilayer (DOPC: DSPE-PEG$_{2000}$ = 90 : 10), simply by mixing the liposome dispersion (prepared in the "Preparation of liposome" section) with the as-prepared PLGA nanoparticle (at a mass ratio of lipid: PLGA = 1 : 2) and then sonicating the resulting mixture in a water bath sonicator (KUDOS, SK5210HP) (at a frequency of 53 kHz and an output power of 100 W for 5 min), followed by centrifugation at 12,000 × $g$ for 15 min to remove excess free liposome. The resulting pellet (i.e., the expected PLGA@lipid nanoparticle) was collected and re-dispersed into Millipore water, which yielded the expected PLGA@lipid stock dispersion.

**Preparation of PEG-absent lecithin-coated nanoparticles.** Lecithin liposome coated nanoparticles were synthetized through similar procedures as used for preparing synthetic lipid bilayer coated nanoparticles (as described in the "Preparations of liposomes, the nanogel@lipid particles, the PLGA@lipid" section) but by replacing the liposome composed of DOPC: DSPE-PEG$_{2000}$ = 90 : 10 with liposome composed of lecithin.

**Characterizations on nanoparticles.** The hydrodynamic diameter and surface zeta-potential (ζ-potential) of a nanoparticle were obtained by monitoring the dispersion of the nanoparticle (at 50 μg/mL in lipid dose in Millipore water) with a nanoparticle analyzer (Nano-ZS90, Malvern) at 25 °C.

Morphology of the PLGA@lipid or PLGA@lec nanoparticle was characterized by negative staining the particle (1 mg/mL in lipid dose in Millipore water) with 1 wt% phosphotungstic acid, adding a few drops of the resulting dispersion onto a copper grid, drying naturally in open air, and then imaging under a transmission electron microscope (TEM) (H-7650, Hitachi) operated at 100 kV. The morphology of a soft nanoparticle (the liposome, or a nanogel@lipid particle) was examined by dispersing the nanoparticle into Millipore water (to a final concentration of 1 mg/mL in lipid dose), freezing the resulting dispersion on a copper grid, and then imaging under a Cryo-TEM (Tecnai G2 spirit 120 kV, Thermo FEI).

**The stability of the lipid bilayer coating on nanoparticles.** To examine the stability of the lipid bilayer shell, we used 188 kPa@lipid and 700 kPa@lipid nanoparticles as the representatives for our model nanoparticles and compared their morphologies under Cryo-electron microscopy (Cryo-EM) before and after a designated treatment. Briefly, on the 1$^{st}$ day after dispersing a nanoparticle (188 kPa@lipid or 700 kPa@lipid, 1 mg/mL in lipid dose) into phosphate buffered saline (PBS), we were characterized particles in the resulting dispersion under Cryo-EM (i.e., without any other treatment), which yielded the morphologies of our intact pristine nanoparticles (i.e., the negative control). To obtain the morphologies of our nanoparticles after their lipid bilayer coating was completely peeled off (i.e., the positive control), we on the 1$^{st}$ day after dispersing a nanoparticle (188 kPa@lipid or

700 kPa@lipid) into PBS treated the resulting dispersion with Triton X-100 (10 μL per 1 mL of nanoparticle dispersion), a surfactant known to be able to peel off a particle's lipid bilayer coating, and then characterized the resulting particles under Cryo-EM. To simulate the conditions a nanoparticle is to encounter both in vitro and in vivo, we incubated our nanoparticles (188 kPa@lipid or 700 kPa@lipid, 1 mg/mL in lipid dose) in fetal bovine serum (FBS)-supplemented phosphate buffered saline (PBS) (v./v., 50%) for 7 consecutive days and then centrifuged the resultant dispersion at 150,000 × $g$ at 4 °C for 1 h to collect the as-treated nanoparticles (after discarding the resulting supernatant). The resulting particles were subsequently washed once with Millipore water centrifugation at 150,000 × $g$ at 4 °C for 1 h, redispersed into PBS (1 mg/mL), and then characterized under Cryo-EM, which yielded the morphologies of our nanoparticles after incubation in protein-present environment for an extended duration.

**Elastic moduli of bulk hydrogels.** The elastic modulus of a nanogel@lipid particle was indirectly indicated by monitoring that of its corresponding bulk hydrogel that was prepared with a same hydrogel-precursor solution (as described in the "Preparations of the nanogel@lipid particles" section). Briefly, a bulk hydrogel was prepared by subjecting a pre-specified hydrogel-precursor solution in a cylinderical glass tube (6 mm in height, 10 mm in diameter) to 40-min irradiation with a UV laser (at 365 nm). To obtain the Young's modulus of the resulting hydrogel (6 mm in height, 10 mm in diameter), the hydrogel was placed in between a pair of parallel-plate compression clamps of a universal testing machine (UTM2502, Suns) (Shenzhen, China) and then subjected to compression with a force ramp at a strain rate of 1 mm/min, which yielded the relationship of compressional force versus deformation. In the as-obtained plot of compressional force versus deformation, the slope equals the Young's modulus of the hydrogel.

**Preparation of mouse plasma.** All animal experiments were conducted in compliance with the guidelines for the care and use of research animals established by the Animal Care and Use Committee at the University of Science and Technology of China. The animals used in this work were ICR (Institute for Cancer Research) (CD-1) mice (female, 1–8 weeks old). Mice were housed at a temperature of 22–25 °C and a 12 h/12 h dark/light cycle. Mouse plasma was collected from female ICR (CD-1) mice. Specifically, twelve female ICR (CD-1) mice (1–8 weeks old) were purchased from Beijing Vital River Laboratory Animal Technology Co., Ltd. and housed in the Animal Research Center at USTC. Mouse blood was collected from the eye pits of the ICR mice once every 15 or more days into a 1-mL centrifuge tube containing heparin sodium powder that was pre-dried from 10 μL of 100 U heparin sodium injection (Changzhou Qianhong Biopharma Co., Ltd. China), stored temporarily on ice, and then centrifuged at 1500 × $g$ for 15 min at 4 °C (5417 R, Eppendorf) to pellet the red and white cells. The supernatant, which is the mouse plasma, was collected and then separated into aliquots (1 mL per aliquot). If the subsequent experiments that need use mouse plasma were ready, some aliquots of the as-collected mouse plasma were randomly selected and used directly right after, while the leftover aliquots were stored at −80 °C prior to use; if the subsequent experiments that need use mouse plasma were not ready, all aliquots of the as-collected mouse plasma were stored at −80 °C prior to use. All aliquots from a same batch of mouse plasma were used within 2 months from the day of their preparation; if there was any aliquot from a same batch of mouse plasma left after 2 months from the day of its preparation, it would be discarded. It should be noted that, prior to use, an aliquot of the as-prepared mouse plasma was taken out of the −80 °C freezer, thawed naturally at room temperature (this step normally takes ~20 min), and then centrifuged at 20,000 × $g$ for 15 min at 4 °C (5417R, Eppendorf) to remove aggregated proteins if there is any.

**Preparations of dye-labeled nanoparticles.** To prepare the dye-labeled nanoparticles, stock dispersions of liposome, nanogel@lipid particles, and PLGA@lipid, which were prepared respectively as described in above sections "Preparation of liposome", "Preparations of the nanogel@lipid particles", and "Preparation of the PLGA@lipid nanoparticle", were diluted into physiological isotonic solutions, followed by mixing with DiD (for pharmacokinetics studies) or Dil (for cellular studies), two membrane dyes, to label the lipid bilayer with the membrane dye. Specifically, for the liposome and the nanogel@lipid particles, the nanoparticle stock dispersion was diluted with 10×PBS buffer solution at a volume ratio of

$10 \times$ PBS: nanoparticle stock dispersion = 10: 90 to a final concentration of 1 mg/mL in lipid dose, which yielded the nanoparticle dispersion in $1 \times$ PBS buffer solution; for the PLGA@lipid, the PLGA@lipid stock dispersion (in Millipore water) was diluted with 50% glucose solution at a volume ratio of 50% glucose: dispersion = 10: 90 to a final concentration of 1 mg/mL in lipid dose, which yielded the PLGA@lipid dispersion in 5% glucose solution. Into a resulting nanoparticle dispersion in physiological isotonic solutions was added a DiD or DiI solution (2 mg/mL in ethanol) at a mass ration of dye: lipid = 1: 100, followed by shaking at room temperature for overnight (~12 h), which yielded the expected dye-labeled nanoparticle. The resulting dye-labeled nanoparticle stock dispersions were stored at 4 °C prior to use.

To quantify the nanoparticle doses in subsequent pharmacokinetics studies and cellular uptake assays which relies on fluorescence signals from DiD and DiI, we firstly monitored the calibration curves of these dye-labeled nanoparticles. Briefly, for a dye-labeled nanoparticle, its stock dispersion as prepared above was diluted serially into mouse plasma (for pharmacokinetics studies) or RIPA Lysis buffer (Beyotime, China) (for cellular uptake assays) to a serial of final concentrations, and the fluorescence intensity of the resulting dilutions were measured with a Spectrum Imaging System (IVIS, Perkin Elmer) for DiD ($\lambda_{ex}/\lambda_{em} = 640$ nm/680 nm) and a fluorimeter (F-4600 spectrofluorometer, Hitachi) for DiI ($\lambda_{ex}/\lambda_{em} = 515$ nm/ 520–750 nm; with slit-widths of 10 nm and 10 nm for excitation and emission wavelengths, respectively), which yielded the relationship of fluorescence emission intensity versus nanoparticle concentration in lipid dose for the dye-labeled nanoparticle.

**Pharmacokinetics studies.** All animal experiments were conducted in compliance with the guidelines for the care and use of research animals established by the Animal Care and Use Committee at the University of Science and Technology of China. Pharmacokinetics of a nanoparticle was assessed by intravenously injecting its corresponding DiD-labeled nanoparticle (prepared as described in the "Preparations of dye-labeled nanoparticles" section) into mouse models and then measuring the blood retentions of the dye at differing time-points after injection. Briefly, twenty-seven female ICR (CD-1) mice (1–8 weeks old) (Beijing Vital River Laboratory Animal Technology Co., Ltd.) were randomly divided into nine groups ($n = 3$ per group), with each group to be subsequently treated with one of the nine DiD-labeled nanoparticles above (i.e., the liposome, the seven nanogel@lipid particles that differ in elasticity, and the PLGA@lipid). Into each mouse was injected (through the tail vein) with the dispersion of a DiD-labeled nanoparticle (200 µL, 200 µg in dose of lipid over nanoparticle). At different time-points after injection (which here are 5 min, 30 min, and 1-, 2-, 4-, 8-, 12-, 24-, 48-h), 100 µL of blood was collected from the eyepit of a mouse, transferred into a centrifuge tube containing heparin sodium powder that was pre-dried from 10 µL of 100 U heparin sodium injection (Changzhou Qianhong Biopharma Co., Ltd. China), stored temporarily on ice, and then centrifuged at $1500 \times g$ for 15 min at 4 °C (5417R, eppendorf). Thirty microliters of the resulting plasma were then transferred into a well of a black 96-well microplate (JingAn Biological, China), and the as-inoculated microplate was subsequently imaged with a Spectrum Imaging System (IVIS, Perkin Elmer) for DiD fluorescence intensity ($\lambda_{ex}/\lambda_{em} = 640$ nm/680 nm), which yielded the concentrations of DiD in blood at different time-points after injection and, in combination with the corresponding calibration curve of fluorescence intensity versus nanoparticle concentration (obtained in the "Preparations of dye-labeled nanoparticles" section) for the DiD-labeled nanoparticle, provided the blood retentions of the nanoparticle in mice. Based on the resulting blood retentions of a DiD-labeled nanoparticle, the blood circulation half-life of the nanoparticle was calculated with a two-compartment model using PKSolver, a freely available menu-driven add-in program for Microsoft Excel. The calculation formula is:

$$C = Ae^{\alpha t} + Be^{\beta t}, \tag{1}$$

C is the concentration of the particle in the plasma, and α and β are rate constants for the distribution and elimination processes, respectively.

**Cellular uptake assays using adherent cells.** DiI-labeled nanoparticles were used to quantify the cellular uptake efficiency of the nanoparticles. Murine macrophage Raw264.7 and Ana-1 cells were used as the representatives for mammalian phagocytes. Briefly, in a tissue culture dish (664970., Greiner Bio-one), RAW 264.7 macrophage cells were cultured in FBS-supplemented DMEM (10%, v./v.) to ~80% confluency, followed by collection with trypsin digestion, centrifugation (800 g for 3 min), and then re-dispersion into FBS-supplemented DMEM to a final cell number density of $6 \times 10^6$/mL. The resulting RAW 264.7 cells were seeded into a 24-well microplate at a cell number density of ~$10^5$ per well and in 0.5 mL FBS or mouse plasma-supplemented DMEM per well, followed by incubation (5% CO₂, at 37 °C) for overnight (~10 h). Into each well of the as-inoculated microplate was then added the dispersion of a DiI-labeled nanoparticle (10 µL,1 mg/mL in lipid dose, prepared as described in the "Preparations of dye-labeled nanoparticles" section), followed by incubation at 37 °C (5% CO₂) for 12 h. The as-treated cells were subsequently washed twice with sterile PBS, followed by 1-h treatment with RIPA Cell Lysis Buffer (0.3 mL) (Beyotime, China) to lyse the cells, and the fluorescence emission spectra of DiI in the resultant dispersions was recorded with a fluorimeter (F-4600 spectrofluorometer, Hitachi) ($\lambda_{ex}./\lambda_{em}. = 515$ nm/ 520–750 nm; with slit-widths of 10 nm and 10 nm for excitation and emission

wavelengths, respectively). From the resulting fluorescence emission spectra were extracted DiI's fluorescence intensity at 570 nm which, combined with the corresponding calibration curve for the DiI-labeled nanoparticle (as obtained in the "Preparations of dye-labeled nanoparticles" section) and the total volume of the dispersion in a well of the microplate, indicated the mass of internalized nanoparticles by the inoculated RAW264.7 cells. A nanoparticle's cellular uptake efficiency is indicated by $M_{uptake}/M \times 100\%$, where $M_{uptake}$ is the mass of internalized nanoparticles by the inoculated cells in a well while M is the mass of total nanoparticles added into the well. Each assay was carried out in triplicate, and the reported results are averages of two independent trials.

Similar procedure was performed to characterize the uptake efficiency of a DiI-labeled nanoparticle by murine macrophage Ana-1 cell, our second model for adherent cells.

**Cellular uptake assays using suspension cells.** Cells in blood are in suspension, rather than adherent. Murine macrophage Ana-1 cell is semi adherent and semi suspended cells. To better simulate cells in blood, when macrophage Ana-1 cells were cultured in FBS-supplemented RMPI 1640 (v./v.: 10%), the suspended cells were collected and used as the representatives for suspension cells. Macrophage Ana-1 cells were cultured in FBS-supplemented RMPI 1640 (v./v.: 10%) (Hyclone) in a tissue culture dish (37 °C, 5% CO₂) to ~70% confluency consistently on a few randomly selected focal planes under a microscopy, followed by collection by centrifugation (800 × g for 3 min) and then re-dispersion into fresh FBS-supplemented RMPI 1640 to a final cell number density of ~$1 \times 10^6$/mL. Then resulting cells were seeded into a 24-well microplate at a cell number density of ~$10^5$ per well and in fresh FBS-supplemented RMPI 1640 (0.5 mL per well). Into each well of the as-seeded microplate was added the dispersion of a DiI-labeled nanoparticle (10 µL,1 mg/mL in lipid dose, prepared as described in the "Preparations of dye-labeled nanoparticles" section), followed by incubation at 37 °C (5% CO₂) with shaking (100 rpm) for 12 h. The as-treated cells were then centrifuged at $1500 \times g$ at 4 °C for 10 min to remove free particles (by discarding the resultant supernatant), followed by washing twice with sterile PBS and then 1-h treatment at room temperature by 0.3 mL of RIPA Lysis Buffer (Beyotime, China) to lyse the cells, and the fluorescence emission spectra of DiI in the resultant dispersions was recorded with a fluorimeter (F-4600 spectrofluorometer, Hitachi) ($\lambda_{ex}/\lambda_{em} = 515$ nm/520–750 nm; with slit-widths of 10 nm and 10 nm for excitation and emission wavelengths, respectively). From the resulting fluorescence emission spectra were extracted DiI's fluorescence intensity at 570 nm, which, combined with the corresponding calibration curve for the DiI-labeled nanoparticle (as obtained in the "Preparations of dye-labeled nanoparticles" section) and the total volume of the dispersion in a well of the microplate, indicated the mass of internalized nanoparticles by the inoculated Ana-1 cells. A nanoparticle's cellular uptake efficiency is indicated by $M_{uptake}/M \times 100\%$, where $M_{uptake}$ is the mass of internalized nanoparticles by the inoculated cells in a well while M is the mass of total nanoparticles added into the well. Each assay was carried out in triplicate, and the reported results are averages of two independent trials.

To further simulate the physiological environment a nanoparticle encounters after entering blood and characterize its cellular uptake efficiency by cells therein, we replaced FBS with mouse plasma (prepared in the "Preparation of mouse plasma" section) and carried out similar procedure as described above but in RMPI 1640 (v./v.: 10%) supplemented with mouse plasma.

**Preparation of nanoparticle-protein complexes.** The preparation of nanoparticle-protein complexes involved the stock dispersions of nine nanoparticles, which are the liposome, the nanogel@lipid particles, PLGA@lipid, and lecithin liposome coated nanoparticles prepared as described in above sections "Preparation of liposome", "Preparations of the nanogel@lipid particles", "Preparation of the PLGA@lipid nanoparticle", "Preparation of PEG-absent lecithin-coated nanoparticles" respectively.

To prepare a nanoparticle-protein complex, stock dispersion of a nanoparticle (~200–400 µL in Millipore water) was diluted into physiological isotonic solutions (to a final nanoparticle dose of 1 mg in lipids and a final total volume of 0.6 mL) either with $10 \times$ PBS (product no. ST476, Beyotime) if the nanoparticle was our liposome or a nanogel@lipid particle and with 50% glucose solution if the nanoparticle was the PLGA@lipid or PLGA@lec, followed by mixing the resulting dilution with the mouse plasma (0.4 mL, prepared as in the "Preparation of mouse plasma" section above). The resulting mixture was subsequently subjected to incubation at 37 °C with shaking (300 rpm) for 12 h, followed by centrifugation at 150,000 g at 4 °C for 2 h (Optima MAX-XP, Beckman Coulter). The resulting supernatant was discarded, and the resulting pellet was collected by washing for three times with Millipore water, which yielded the expected nanoparticle-protein complexes that were used immediately and directly (i.e., without further storage or any other treatment) in the subsequent experiments (which include "Measurement of the total amount of adsorbed proteins", "Sodium Dodecyl Sulfate-Polyacrylamide Gel Electrophoresis (SDS-PAGE) analysis", and "Liquid-chromatography mass-spectrometry (LC-MS) analysis").

**Measurement of the total amount of adsorbed proteins.** The total amount of proteins adsorbed on a nanoparticle was measured by using a Bradford Protein

Assay Kit (Beyotime, China), which indicates protein content with the absorbance at 595 nm due to the binding of Coomassie Blue G-250 with proteins and includes a standard bovine serum albumin (BSA) solution (at 5 mg/mL) for monitoring the calibration curve of absorbance at 595 nm versus BSA concentration. Briefly, the standard BSA solution (5 mg/mL) was diluted serially into Millipore water to a serial of final concentrations (specifically, at 1.5, 1, 0.75, 0.5, 0.25, and 0.125 mg/mL). The resulting dilutions (5 μL per well) were subsequently added into a 96-well microplate, followed by addition of Coomassie Blue G-250 staining solution (250 μL per well). The as-set microplate was then placed into a microplate reader (Varioskan® Flash Spectral Scanning Multimode Readers, ThermoFisher) to record the absorbance at 595 nm. Millipore water (i.e., 0 mg/mL BSA) was included as a blank control. Plot of the resulting absorbance (at 595 nm) versus protein concentration yielded the calibration curve for BSA, a model for proteins.

A particle-protein complex (prepared as described in the "Preparation of particle-protein complexes" section) was re-dispersed into Millipore water (100 μL). The resulting dispersion of the nanoparticle-protein complex was then added into a 96-well microplate (5 μL per well), followed by addition of Comassie Blue G-250 staining solution (250 μL per well). The as-set microplate was subsequently placed into a microplate reader (Varioskan® Flash Spectral Scanning Multimode Readers, Thermo fisher) to record the absorbance at 595 nm. The as-measured absorbance at 595 nm, in combination with the calibration curve of absorbance (595 nm) versus BSA concentration measured in the same trial, yielded the concentration of proteins in a well of the microplate, which combined with the final volume of the particle-protein complex dispersion provided the total amount of proteins adsorbed on the nanoparticle. The reported results are averages of three independent trials.

**Sodium dodecyl sulfate-polyacrylamide gel electrophoresis.** One-dimensional SDS-PAGE was performed under reducing conditions in 10% of acrylamide, according to a previously reported protocol[63,64]. The 10% SDS-PAGE gel was prepared firstly. Two glass plates were clamped in the casting frames, appropriate amount of separating gel solution (product no. P0012A, Beyotime, China) (preparing 10 mL 10% separating gel solution need add 2.7 mL Millipore water, 3.3 mL 30% Acr-Bis (Acr: Bis ∼ 29:1) solution, 3.8 mL Tris (1 M, pH 8.8), 0.1 mL 10% SDS solution, 0.1 mL 10% ammonium persulfate solution, and 4 μL tetramethylethylenediamine) was pipetted into the gap between the glass plates to a level which will allow the comb to be inserted with 5 mm between the bottom of the wells and the top of the separating gel. Millipore water (0.4 mL) was then added to cover the separating gel solution to ensure a flat interface between the seperating and stacking gels, and the gel solution was allowed to polymerize for 30 min. After discarding the water, the separating gel was still left, the stacking gel solution (product no. P0012A, Beyotime, China) (preparing 4 mL stacking gel solution need add 2.7 mL Millipore water, 0.67 mL 30%Acr-Bis (29:1) solution, 0.5 mL 1 M Tris (pH 6.8), 0.04 mL 10% SDS solution, 0.04 mL 10% ammonium persulfate solution and 4 μL tetra-methylethylenediamine) was then pipetted into the gap until a overflow. And the well-forming comb was inserted into the gel solution without trapping air under the teeth. The comb was taken out with a complete gelation of the stacking gel solution after ∼30 min polymerization, which yield the expected SDS-PAGE gel.

A particle-protein complex (prepared as described in "Preparation of particle-protein complexes" section) was re-dispersed into Millipore water (50 μL). The resulting dispersion (10 μL) was subsequently mixed with the 2× SDS-PAGE loading buffer (Beyotime) (10 μL) to solubilize the nanoparticle-associated proteins and then incubated in a hot water bath (at 95 °C) for 5–10 min to denature the proteins. The resulting protein-containing loading buffer was loaded onto a channel of the gel, and electrophoresis was subsequently performed (VE 680, Tanon) at 120 V for 150 min in the Tris-Gly electrophoresis buffer (3.02, 18.8 and 1 g/L in Millipore water for tris base, glycine and SDS, respectively), followed by transferring and immersing the SDS-PAGE gel into a Coomassie blue R250 staining solution (Beyotime) for 2 h to stain the protein bands and then washing in decolorizing solution (methanol/ acetic acid/ ultrapure water = 3: 1: 6 (in volume ratio)) to visualize the protein bands. Photographs of uncropped and unprocessed SDS-PAGE gels are provided in our Source Data file which has been provided as a supplementary file for publication.

**Liquid-chromatography mass-spectrometry (LC-MS) analysis.** A freshly prepared nanoparticle-protein complex (prepared as described in the "Preparation of particle-protein complexes" section) was re-dispersed into NH₄HCO₃ solution (50 mM, in Millipore water), followed by reduction via addition of freshly prepared dithiothreitol (DTT) (to a final DTT concentration of 100 mM) and then incubation at 50 °C for 15 min. The as-reduced dispersion was subsequently cooled naturally (via standing still on a lab bench) to room temperature, followed by alkylation through addition of freshly prepared iodoacetamide (IAM) (to a final IAM concentration of 300 mM) and then incubation at room temperature (via standing still on a lab bench) in dark for 15 min. Into the as-alkylated dispersion was added modified sequencing grade trypsin (at a trypsin to protein weight ratio of 1: 50, by assuming the total protein mass to be 100 μg), and the resulting dispersion was then incubated at 37 °C overnight (16–18 h), which yielded the sample that is to be analyzed with LC-MS but needs to be stored at −20 °C for <14 days due to the queue waiting for LC-MS analysis.

Prior to LC-MS analysis, the as-stored sample was taken out of the −20 °C refrigerator, thawed naturally (via standing still on a lab bench) to room temperature, and then diluted to a protein content of ≥50 μg/mL with 0.1% formic acid prior to characterizations with an LC-MS instrument (Q Exactive Plus, Thermo).

The dissolved peptide sample was then analyzed with a high-performance liquid chromatography (HPLC) system (Thermo scientific, model EASY-nLC 1200 system) coupled with a mass spectrometer (Thermo scientific, model Q Exactive Plus). Tryptic peptides were separated on the EASY-nLC 1200 system equipped with a C18 analytical reversed-phase column (particle size: 2 μm; pore size: 100 Å; diameter × length: 50 μm × 150 mm; Acclaim® PepMap™ RSLC, thermo scientific) and a C18 trap column (particle size: 5 μm; pore size: 100 Å; diameter × length: 100 μm × 20 mm; Acclaim® PepMap™ 100, thermo scientific). The samples were processed with mobile phase solvent A consisting of 0.1% (v/v) formic acid in pure water, and mobile phase solvent B was 80% acetonitrile with 0.1% (v/v) formic acid in water. The separation was performed over 100 min using a gradient of 3–35% solvent B at a sample flow rate of 0.3 μl/min. Typical sample injection volume was 1 μL.

The high-performance liquid chromatography (HPLC) system was on-line coupled with a mass spectrometer (Thermo scientific, model Q Exactive Plus) which was controlled by the mass spectrometry (MS) instrument-control software Thermo Xcalibur (version: 4.0), and the mass spectrometer was set at the data-dependent mode to acquire MS/MS data. Electrospray ionization (ESI) was performed in positive ion mode. The ionization voltage was 2 kV. The capillary temperature was set to 320 °C. The normalized collision energy was set 27%, and the default charge state was at 2. Data were acquired within a range of m/z 150–2000 Da in one full scan.

The mass spectrometric data were then used to search against the UniProt protein database with Thermo Proteome Discoverer software suite (version: 2.2.0.388). During database searches, the protein strict and relaxed false discovery rates were set to 1% and 5%, respectively. The mass tolerances for precursor mass and fragment mass were 10 ppm and 0.02 Da, respectively. The following criteria were used for the search: one missing cleavage, fixed carbamidomethyl modification of cysteine, and variable oxidation of methionine.

The relative abundance of a specific protein in the corona of a nanoparticle was determined through the method of spectral counting (SpC)[52], which represents the total number of the MS/MS spectra for all peptides attributed to a matched protein. For each protein identified in a corona, its SpC was normalized to the protein mass in the corona through the following equation[41,65] and expressed as the relative abundance in corona:

$$NpSpC_k = \left\{ \frac{(SpC/(M_w)_k)}{\sum_{t=1}^{n}(SpC/(M_w)_t)} \right\} \times 100 \qquad (2)$$

where $NpSpC_k$ is the normalized percentage of the spectral count for protein k, SpC is the spectral count for a protein identified in the corona, and Mw is the molecular weight (in kDa) for a protein identified in the corona, n is the total number for proteins identified in the corona, and k is the label for the protein of interest. This correction is based on the protein size and evaluates the relative contribution of each protein present in the corona of nanoparticles[66].

**Study of the effect of ApoA1 on cellular uptake.** To study the effect of ApoA1 on cellular uptake, Murine macrophage suspended and adherent Ana-1 cell was used as a representative for mammalian phagocytes. We replaced ApoA1 protein solution with mouse plasma or FBS and carried out similar procedure as described in "Cellular uptake assays using suspension cells" and "Cellular uptake assays using adherent cells" above but in RMPI 1640 or DMEM (v./v.: 10%) supplemented with ApoA1 protein solution (25 μL, 1 mg/mL in 10 mM NH₄HCO₃, into per well). Blank and negative controls are cells treated similarly but with PBS (25 μL, per well) and BSA (25 μL, 1 mg/mL in PBS, into per well) as substituents for the ApoA1 protein solution, respectively.

**Isothermal Titration Calorimetry (ITC) analysis.** The ITC measurements were performed using a MicroCal PEAQ-ITC (Malvern) with an effective cell volume of 200 μL. The dispersion of a nanoparticle (0.5 mg/mL liposome, in lipid dose) in 10 mM ammonium bicarbonate solution was added into the sample cell which was kept at 25 °C and stirred at 750 rpm, while the ApoA1 solution (40 μL, 0.4 mg/mL in 10 mM ammonium bicarbonate solution) was added as the injectant into the syringe. The ApoA1 solution was then titrated via injection into the nanoparticle dispersion, with the volume of each injection kept constant at 2 μL while the spacing between injections constant at 120 s.

**Effects of ApoA1 antibody on cellular uptake.** The three kinds of freshly prepared nanoparticle-protein complexes (namely, liposome-protein complex, 188 kPa@lipid-protein complex, PLGA@lipid-protein complex) (prepared as described in the "Preparation of particle-protein complexes" section) were re-dispersed into PBS (to a final concentration of 5 μg/μL in lipid dose). Then, the nanoparticle-protein complexes were incubated with ApoA1 antibody (1 μg/μL in 0.01 M TBS (pH 7.4) with 1% BSA, 0.03% Proclin300 and 50% Glycerol) (bs-4573R, Bioss) (Beijing, China) for 1 h (at a volume ratio of the nanoparticle-protein complex suspension: antibody solution ∼ 5: 1), and the control group (i.e., the nanoparticle-protein complexes without any treatment) was incubated with PBS for 1 h. 6 μL of the resulting mixture

of nanoparticle-protein complex suspension and ApoA1 antibody was added into each well of a 24-well microplate where the adherent Ana-1 were pre-seeded at a cell number density of ~$10^5$ per well. The cellular uptake assays of the adherent Ana-1 were then carried out in a similar procedure as described in "Cellular uptake assays using adherent cells".

**Effects of ApoA1 antibody on pharmacokinetics**. The three kinds of freshly prepared nanoparticle-protein complexes (namely, liposome-protein complex, 188 kPa@lipid-protein complex, PLGA@lipid-protein complex) (prepared as described in the "Preparation of particle-protein complexes" section) was re-dispersed into PBS (to a final concentration of 1 mg/mL in lipid dose). The nanoparticle-protein complex suspension was incubated with ApoA1 antibody (1 μg/μL) for 1 h (at a volume ratio of the nanoparticle-protein complex suspension: antibody solution ~ 24: 1), which yielded the corresponding mixture with ApoA1 antibody for the nanoparticle-protein complex. Thirty-six female ICR (CD-1) mice (1–8 weeks old) (Beijing Vital River Laboratory Animal Technology Co., Ltd.) were randomly divided into six groups ($n = 6$ mice per group) and each mouse was intravenously injected through tail vein with either a nanoparticle-protein complex suspension (200 μL per mouse) or its corresponding mixture with ApoA1 antibody (200 μL per mouse). Then the similar procedure as described in "Pharmacokinetics studies" was carried out.

**Calculations on Pearson's correlation coefficient r**. Pearson correlation coefficient r (i.e., Pearson's r) can be any value between −1 and 1 depending on the extent of collinearity. For our nanoparticles, the Pearson's correlation coefficient r between the relative abundance of a specific protein in corona versus blood clearance half-life was calculated using OriginPro (Version 2018), to determine whether the adsorption of the specific protein on our nanoparticles correlates strongly with our nanoparticles' system circulation. In these calculations, the blood clearance half-life of a nanoparticle was averages of two independent trials.

**Reporting summary**. Further information on research design is available in the Nature Research Reporting Summary linked to this article.

## Data availability

The authors declare that data supporting the findings of this study are available within the paper and its Supplementary Information files. Source data are provided with this paper. Source data are available for Figs. 1e, 2b, c, e–i, k, 3a–h, j, 4a–d, f, 5d–i and Supplementary Figs. 2b, 3, 4b, c, 7, 8, 10–13, 15–17, 18b, c, 19, 20 in the associated source data file. The mass spectrometry proteomics data have been deposited to the ProteomeXchange Consortium (http://proteomecentral.proteomexchange.org) via the iProX partner repository [1] with the dataset identifier PXD034004. [1] Ma J, et al. (2019) iProX: an integrated proteome resource. Nucleic Acids Res, 47, D1211-D1217. Source data are provided with this paper.

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

## Acknowledgements

We gratefully thank Professors Yangzhong Liu, Zhigang Wang and Yucai Wang at USTC for use of his facilities and Gao Wu from the Core Facility Center for Life Sciences of USTC for helping perform the LC-MS characterizations. This work was supported in part by the National Natural Science Foundation of China (12174366 and 31671014 (L.Y.)), the Ministry of Education of China (via the Fundamental Research Funds for the Central Universities) (WK3450000005 (L.Y.)), and Anhui Provincial Natural Science Foundation (2108085MC93 (L.Y.)).

## Author contributions

L.Y. conceived the idea; M.L., T.L., F.F., and F. G. performed the experiments; L. Y. and M.L. analyzed the results; L.Y., M.L., T.L., X.J., and S.C. wrote the paper.

## Competing interests

The authors declare no competing interests.
