## [Peer Review File · Nature Communications]

Nanoparticle Elasticity Affects Systemic Circulation Lifetime by Modulating Adsorption of Apolipoprotein A-I in Corona FormationREVIEWER COMMENTS

Reviewer #1 (Remarks to the Author):

This paper investigated the effect of nanoparticle elasticity on their systemic circulation, and concluded that apolipoprotein A-I in corona formation was the only major corona protein whose relative abundance in corona correlated strongly with the NPs systemic circulation in mice. Firstly, a nanoparticle library with different elasticity 45 kPa - 760 MPa was fabricated consisting of liposomes, hydrogel@lipid and PLGA@lipid. The effect of NP elasticity on protein corona was investigated. Many interesting findings were reported, but more in an observational way.

(1) The nanoparticles in the library actually have quite different properties, for example size and surface charge, especially the charge. The authors claimed that they fabricated core-shell nanoparticles with the same PEGylated lipid bilayer, if this is the case, they should have very similar zeta potential, but they are very different ranging from about -15 to -30 mV. So the PEG density on the lipid bilayer should be carefully characterised to ensure they really have the same surface properties, otherwise, the difference in surface charges should be taken into consideration.

(2) The stability of the lipid bilayer on the particle surface should be characterised, as the following in vitro and in vitro experiments, dyes were loaded into the lipid bilayers, if they are easy to be peeled off from the particles, what was measured would not be the nanoparticles.

(3) Some very interesting but contradictive results were shown in Fig 2. Liposomes as the most soft systems showed the shortest circulation time and highest cellular uptake in RAW macrophage cells, so there is no direct correlation between the particle stiffness and their circulation if liposomes are considered as the softest system.

(4) The conclusion about the correlation between the relative abundance apolipoprotein A-I in corona and their circulation time is bold, as shown in Fig. 3e, the adsorption of other proteins such as complement proteins varies a lot with the particle elasticity. So the analysis of the correlation between the protein corona and circulation time is too simplified, no clear trend doesn't mean no contribution.

(5) The fundamental understanding of the role of apolipoprotein A-I in protein corona and consequently in controlling circulation time is lacking. There are many studies about protein corona and the role of apolipoprotein A-I in controlling NP biological identities. More in depth discussion should be included.

Reviewer #2 (Remarks to the Author):

In their manuscript entitled "Nanoparticle Elasticity Affects Systemic Circulation by Modulating the Adsorption of Apolipoprotein A-I in Corona Formation" the authors investigate the effects of nanoparticle core density on the protein corona and systemic circulation.

The study seems overall well designed and results are clearly presented. The study could be an important contribution to the field, as it could provide a basis for the yet unknown mechanisms why nanoparticles of different elasticity show different biological behaviours (e.g. uptake, circulation etc).

However, I have one major point that requires to be addressed in a revision:

A key aspect of the study is the investigation of the protein corona and its characterization by LC-MS based proteomics.

Unfortunately, the proteomics data are very insufficiently described. No data are provided except on a highly aggregated level (i.e. figures).

It is not even clear if the experiments or analyses were performed in replicates, or if data from an n=1 proteomic analysis are shown. As for all experiments, multiple biological replicates are required. I also highly recommend the analysis of multiple technical replicates to achieve reliable quantification values using a spectral counting-based quantification approach.

Unfortunately, the authors also chose not to present any quantification data, making it completely impossible to judge the validity of proteomic results.

The authors must provide the entire proteomics datasets (identification and quantification results) in the supplementary information. Data should include (at least): Protein Identifier, Scores(peptide and protein level), FDR, number of peptides per proteins, quantification values from all biological and technical replicates.

The methods for proteomic analysis are very insufficiently described.

- No details at all regarding LC conditions (e.g. instrument, column, solvents, flow rates, gradients, temperature, injection amount per replicate....) are provided.

- No details at all on instrument settings or acquisition settings are provided.

- Not even the most basic details for database search parameters (name and version of database search engine, name and version of database, mass tolerances, FDR, modifications, etc) are provided.

Please refer to MIAPE guidelines for a proper reporting of proteomics datasets.

Additionally, the proteomic datasets (rawdata and search results) need to be submitted to a public repository (i.e. ProteomeXchange) to allow other researchers to access the data.

Response to Reviewer #1 (Remarks to the Author):

This paper investigated the effect of nanoparticle elasticity on their systemic circulation, and concluded that apolipoprotein A-I in corona formation was the only major corona protein whose relative abundance in corona correlated strongly with the NPs systemic circulation in mice. Firstly, a nanoparticle library with different elasticity 45 kPa - 760 MPa was fabricated consisting of liposomes, hydrogel@lipid and PLGA@lipid. The effect of NP elasticity on protein corona was investigated. Many interesting findings were reported, but more in an observational way.

(1) The nanoparticles in the library actually have quite different properties, for example size and surface charge, especially the charge. The authors claimed that they fabricated core-shell nanoparticles with the same PEGylated lipid bilayer, if this is the case, they should have very similar zeta potential, but they are very different ranging from about -15 to -30 mV. So the PEG density on the lipid bilayer should be carefully characterised to ensure they really have the same surface properties, otherwise, the difference in surface charges should be taken into consideration.

Author Response: We thank the reviewer for the critical comments. As pointed out by the reviewer, our nanoparticles, though unanimously negative in zeta-potential, exhibited differing readings of zeta-potential in the corresponding experiments. Nevertheless, particles with a same surface chemistry (including charge) may differ in zeta-potential, for reasons as follow.

When a particle is dispersed into an aqueous environment like water (Figure R1), an electrical double layer (*i.e.*, the stern layer and the diffuse layer) is formed at the particle-liquid interface (Park, S.-J.; Seo, M.-K., *Interface Science and Technology: Intermolecular Force* Ch.1, Elsevier. Press, **2011**). Within the diffuse layer, there is a notional boundary (*i.e.*, the slipping plane) which surrounds the particle and separates it from the external environment: When the particle moves, ions internal to this boundary move together with it whereas those external to this boundary stay with the bulk environment. The potential at this notional boundary (*i.e.*, surface of hydrodynamic shear) is the zeta-potential of the particle (Figure R1). Clearly, experimental zeta-potential measurement of a particle doesn't directly measure the absolute charges on the outermost layer of the particle, but instead the potential of this notional boundary (*i.e.*, the slipping plane).

Figure R1. Schematic representation of zeta-potential.

For a core-shell structured particle, the core of the particle may partially screen the charges on the outermost layer of the particle (*i.e.*, the outer-surface of its shell) and consequently affect the ion adsorption within the electrical double layer surrounding the particle — which in return determines its zeta-potential. In another word, core-shell structured particles of a same shell (*i.e.*, same surface chemistry) but different cores may exhibit different zeta-potentials (Supplementary Figure 5) due to the contribution of the particle core. Indeed, supportive evidences to this are found in our literature research on the reported zeta-potentials of lipid or cellular membrane-coated nanoparticles (Supplementary Table 6); as our model nanoparticles in this work are liposomes and lipid bilayer-coated particles, we confined our literature research to lipid or cellular membrane-coated nanoparticles.

Supplementary Figure 5. The surface chemistry of both lipid or cellular membrane vesicles and lipid or cellular membrane coated nanoparticles are same, but they are different in zeta-potential due to the contribution of the particle core to partially screen the charges on the outermost layer of the particle.

Our literature research results (Supplementary Table 6) revealed that none of the lipid or cellular membrane-coated nanoparticles exhibited a same zeta-potential as did its corresponding shell (*i.e.*, the membrane vesicle used to coat the nanoparticle of interest), despite of their similarity in surface chemistry. Of note, this conclusion retains despite that the lipid or cellular membrane-coated nanoparticles we found contain cores differing significantly in materials (inorganic *versus* organic *versus* bacterial) or shells of distinct compositions (synthetic lipid bilayer of distinct compositions *versus* natural cellular membranes of distinct origins). Moreover, zeta-potential even differs between membrane-coated nanoparticles of a same membrane shell but slightly differing cores (Langmuir, 2010, 26, 12081; Nat. Commun., 2021, 12, 1999; Nat. Commun., 2019, 10, 5783); this is the case no matter whether the cores are inorganic SiO₂ nanoparticles (Langmuir, 2010, 26, 12081), polymeric PLA (polylactic acid) nanoparticles (Nat. Commun., 2021, 12, 1999), or bacterial cells (Nat. Commun., 2019, 10, 5783). Clearly, for a core-shell structured nanoparticle, its shell is not the sole factor that determines its zeta-potential; on this aspect, its core matters as well. In another word, that core-shell structured nanoparticles exhibit different zeta-potentials does not necessarily suggest that these particles must differ in surface chemistry.

Supplementary Table 6. The summary of the surface charge core-shell nanoparticles.

Composition		Zeta-potential (mV)			Ref.
core	shell	core	core-shell NP	shell	
polystyrene NP ^a	DPPC	-58	-40	-3	Langmuir , 2005 , 21, 1305
	DPTAP		22	62	
AuNR ^b	DPPC/DPPG	54.1	27.4	-19.2	Int. J. Nanomed. , 2015 , 2015, 33
AuNP	DPPC/DPTAP	-27.33	-9.87	13.78	Nanomaterials , 2018 , 8, 143
ZnO NP	DOPC	26	1.3	-15	ACS Nano , 2021 , 15, 19756
Fe ₃ O ₄ NP	cancer-erythrocyte hybrid membrane	-31.23	-24.54	-19.28	Colloids Surf., B , 2020 , 188, 110755
PCL-PEG ^c NP	neutrophil membranes	-11.3	-13.6	-17.3	Langmuir , 2010 , 26, 12081
SiO ₂ NP (d: 100nm)	DMPC	-42.1	-22.4	0.082	Nat. Commun. , 2021 , 12, 1999
SiO ₂ NP (d: 40-50nm)		-39.2	-13.3		
PLA ^d NP	platelet membrane	-31.87	-33.03	the same (N.D)	Nat. Commun. , 2019 , 10, 5783
PLA NP-R848 ^e		-28.7	-28.2		
Escherichia coli Nissle 1917	DOPA	-38.4	-28.1	the same (N.D)	Nat. Commun. , 2019 , 10, 5783
S. aureus		-32.4	-19.6		
E. faecalis		-28.8	-17.9		

^a NP denotes nanoparticle; ^b NR denotes nanorod; ^c PCL-PEG denotes polycaprolactone-poly(ethylene glycol) block copolymer; ^d PLA denotes polylactic acid nanoparticle; ^e R848 represents resiquimod; N.D. denotes 'not defined'.

Back to our model nanoparticles in this work, they are core-shell structured nanoparticles, in which the shell is a lipid bilayer while the core varies from water over soft hydrogels to PLGA (poly(lactic-co-glycolic acid)) nano-micelle. In lipid relevant research fields, it is a widely-acknowledged practice that the chemistry of a liposome or lipid bilayer is determined by the lipid composition used for preparing the liposome or lipid bilayer (*Nat. Nanotechnol.*, **2011**, 6, 253; *Drug Delivery Transl. Res.*, **2022**, 12, 647; *Nanomedicine*, **2019**, 17, 82; *J. Controlled Release*, **2005**, 105, 305). Our model nanoparticles (excepting PLGA@lipid and PLGA@lethicin) were prepared by following the standard rehydration technique for liposome preparation except that a slight modification was incorporated for hydrogel@lipid nanoparticles; the liposomes were prepared by following the standard rehydration technique for liposome preparation, and the hydrogel@lipid nanoparticles were prepared through a similar rehydration technique except having replaced ultra-pure water for rehydration with hydrogel precursor solutions of differing monomer to cross-linker relative ratios—which after extrusion and ultra-violet light irradiation in subsequent steps provided lipid bilayer-enclosed hydrogel nanoparticles of differing elasticity. Through this procedure, the resulting liposome (no matter whether their interior content is pure water or a hydrogel precursor solution) offers a self-enclosing lipid bilayer (*i.e.*, the shell) whose chemistry is determined by the lipid composition used for preparing the dehydrated lipid thin film, which after rehydration offers the lipid bilayer shell for the liposome and the hydrogel@lipid nanoparticles. In this work, we used two sets of model nanoparticles, with one set having a lipid bilayer shell with a composition of DOPC: DSPE-PEG₂₀₀₀ = 90:10 (mass ratio) while another set having a lipid bilayer shell composed of lethicin alone (*i.e.*, lethicin at a mass ratio of 100%). Within each set of our model nanoparticles, we correspondingly fixed the lipid composition for preparing the dehydrated lipid thin film either at DOPC: DSPE-PEG₂₀₀₀ = 90:10 or lethicin alone. The two stiffest particles, PLGA@lipid and PLGA @lethicin, were prepared by coating a preformed PLGA nano-micelle with a liposome (composed of either DSPE-PEG₂₀₀₀ = 90: 10 or 100% lethicin), and coating a liposome over a nanoparticle is virtually a process that transfers the self-enclosing lipid bilayer of the liposome as a lipid bilayer shell over the particle (*Adv. Colloid Interface Sci.*, **2007**, 133, 1; *Nano LIFE*, **2010**, 01, 163) and the chemistry of the resulting lipid bilayer shell is determined by the lipid composition of the liposome (*J. Immunol. Methods*, **1995**, 185, 81; *Langmuir*, **2003**, 19, 1654). Therefore, it is reasonable for us to expect that, within a given set of our model particles, all particles should be the same in surface chemistry. Specifically, for all our DOPC: DSPE-PEG₂₀₀₀ = 90:10 bilayer-coated nanoparticles, the surface PEG density, which is determined by the relative content of DSPE-PEG₂₀₀₀, should be similar.

Of course, to unveil the effects of elasticity on nanoparticle fate in physiological environment, it is necessary to dissect the effects of elasticity from those of other physiochemical factors like surface charge and surface PEG density. As stated above, within each given set of our model nanoparticle, the particles are reasonably expected

to be same or similar in surface chemistry, which includes surface charge and surface PEG density if PEGylated lipid was involved. Based on analysis above, we reasonably expect that the reason why our DOPC:DSPE-PEG₂₀₀₀ = 90:10 bilayer-coated nanoparticles exhibited slightly different readings in zeta-potential (Figure 1e) must arise because of their significantly differing cores, which ranges from ultra-pure water (for the liposome) over hydrogel cores of differing elasticity (for the hydrogel@lipid particles) to PLGA nano-micelles (for the PLGA@lipid).

Of note, the zeta-potentials of our DOPC:DSPE-PEG₂₀₀₀ = 90:10 bilayer-coated nanoparticles are unanimously negative (Figure 1e). Using polystyrene latex particles that are increasingly negative in surface charge density as model nanoparticles, a previous study has found that increasing nanoparticle surface charge density increases the total amount of adsorbed proteins but imposes negligible effects on the qualitative and quantitative composition of the adsorbed protein pattern (*Eur. J. Pharm. Biopharm.*, **2002**, 54, 165–170). A similar trend was observed in a review paper (*Chem. Soc. Rev.*, **2012**, 41, 2780–2799) which summarized the influence of nanoparticle surface charge on protein corona for anionic polystyrene nanoparticles (Supplementary Table 7). In contrast, when the zeta-potentials of particles are changed from negative to positive, the adsorbed protein pattern will be significantly changed (*J. Biomed. Mater. Res.*, **2003**, 65, 319–326; *Proc. Natl. Acad. Sci. U. S. A.*, **2008**, 105, 14265). Therefore, for nanoparticles that differ only in zeta-potential, it is the sign of their zeta-potentials, rather than the absolute values of their zeta-potentials, that significantly affect their corona protein compositions.

Supplementary Table 7. Qualitative relationships between changes in nanomaterial surface charge and the parameters of the resulting protein corona.

	Parameters of the protein corona			
	Density/ thickness	Identity/ quantity	Conformational change	Affinity
↑ Surface Charge density	Increase	No change	Increase	Increase

Interestingly, though the zeta-potentials of our DOPC: DSPE-PEG₂₀₀₀ = 90:10 bilayer-coated nanoparticles are unanimously negative (Figure 1e), their members acquired protein coronas of significantly different compositions (Figure 3f), which — together with their similarity in size, shape, and surface chemistry (including PEG density) but significant difference in elasticity — in return stresses the importance of nanoparticle elasticity in protein corona.

(2) *The stability of the lipid bilayer on the particle surface should be characterised, as the following in vitro and in vitro experiments, dyes were loaded into the lipid bilayers, if they are easy to be peeled off from the particles, what was measured would not be the nanoparticles.*

Author Response: We thank the reviewer for the kind suggestion. In revision, we examined the stability of the lipid bilayer over nanoparticle, by using 188kPa@lipid and 700kPa@lipid as the representatives for our model nanoparticles and comparing their morphologies under Cryo-electron microscopy (Cryo-EM) before and after treatment that simulates the conditions our model nanoparticles were to encounter both *in vitro* and *in vivo*.

To construct the control nanoparticles resulted from complete peel-off of the lipid bilayers from our model nanoparticles, we treated 188kPa@lipid and 700kPa@lipid nanoparticles with Triton X-100, as treating lipid bilayer-coated nanoparticles with Triton X-100 is a demonstrated method to peel off their lipid bilayer coating (*Macromolecules*, **2002**, 35, 1911), and then characterized the morphology of the as-treated nanoparticles under Cryo-EM. Moreover, to simulate the conditions our model nanoparticles were to encounter in the *in vitro* cell studies and in the *in vivo* animal studies and to examine whether the lipid bilayer coatings over our model nanoparticles were stable under those conditions, we incubated our nanoparticles in fetal bovine serum (FBS)-supplemented phosphate buffered saline (PBS) (v./v., 50%) for 7 consecutive days, separated the particles from supernatant *via* centrifugation at 150,000 g at 4 °C for 1h, and collected (after discarding the resulting supernatant) the resulting pellet (*i.e.*, the as-incubated particles) by washing with PBS for two times, redispersed the as-collected pellet into PBS, and then characterized the morphologies of the as-incubated nanoparticles under Cryo-EM.

Our results (Supplementary Figure 6) reveal that, on the 1st day after being dispersed into PBS, our pristine 188kPa@lipid and 700kPa@lipid nanoparticles (left column) both exhibited an intact core-shell structure under Cryo-EM, in which a thin shell (thickness < 10 nm) (as indicated by white arrow) is clearly observed over the outermost surface of a nanoparticle of slightly lower electron density contrast (Supplementary Figure 6, left column). In stark contrast, the particles after Triton X-100 treatment (Supplementary Figure 6, middle column) completely lost the core-shell structure characteristic of their pristine precursors (*i.e.*, 188kPa@lipid and 700kPa@lipid particles) but instead appeared to be nanospheres with near uniform electron density contrast yet sizes comparable with those of their pristine precursors. In addition to these nanospheres, there emerged many tiny particles of much smaller sizes (<10 nm) (indicated by red arrow), which are likely attributable to lipid micelles formed by the lipids after their peel-off from the nanoparticle cores by Triton X-100, consistent with what was observed in the dynamic light scattering experiments (Supplementary Figure 2). Interestingly, the resulting particles after 7-day incubation in FBS-supplemented PBS (Supplementary Figure 6, right column) exhibited a core-shell structure, in which a thin shell (thickness < 10 nm) (as indicated by white arrow) is clearly observed over the outermost surface of a nanoparticle of slightly lower electron density contrast, as did their pristine precursors (*i.e.*, 188kPa@lipid and 700kPa@lipid particles) (Supplementary Figure 6, left column). Collectively, these observations indicate that 188kPa@lipid and 700kPa@lipid particles both retained their characteristic core-shell

structure even after 7-day incubation in FBS-supplemented PBS, rather than losing their lipid bilayer shell as observed with the same particles but after Triton X-100 treatment, suggesting that the lipid bilayer coating over our model nanoparticles is stable.

Supplementary Figure 6. Cryo-electron microscope (Cryo-EM) images of our (top) 188kPa@lipid and (bottom) 700kPa@lipid nanoparticles (left) on the 1st day after being dispersed into PBS, (middle) after treatment with Triton X-100, and (right) after 7-day incubation in FBS-supplemented PBS. White arrows indicate the locations of the lipid bilayers, and red arrows indicate small micellar particles composed of lipids peeled off our core-shell structured nanoparticles. (Inset) Schematic illustrations on nanoparticles respectively present in the corresponding dispersion samples. Scale bar = 50 nm.

The stability of the lipid bilayer coating over a nanoparticle is observed not just by us. Consistent with our observations, a prior study reported that lipid bilayer-coated nanoparticles retain their core-shell structure with the lipid bilayer coating remaining intact even after 2-month storage at 4 °C (*Macromolecules*, **2006**, 39, 5885–5890).

(3) Some very interesting but contradictive results were shown in Fig 2. Liposomes as the most soft systems showed the shortest circulation time and highest cellular uptake in RAW macrophage cells, so there is no direct correlation between the particle stiffness and their circulation if liposomes are considered as the softest system.

Author Response: We thank the reviewer for the critical comments. We agree with the reviewer that, among our model nanoparticles, the empty liposome (DOPC: DSPE-PEG₂₀₀₀ = 90:10) (*i.e.*, the softest member of our model PEGylated nanoparticle family) has quite high cellular uptake efficiency (Figure 2e-h and k). When incubated with adherent macrophage cells which are good models for simulating resident macrophages

in organs like liver, our liposome exhibited the second-highest average cellular uptake by both RAW 264.7 and Ana-1 cells in FBS-supplemented DMEM media (Figure 2e-f) while the fourth- and fifth-highest cellular uptake respectively by RAW 264.7 and Ana-1 cells in mouse plasma-supplemented DMEM media (Figure 2g-h). When incubated with suspended macrophage cells which are good models for simulating cells in blood like circulating phagocytes, our liposome exhibited the highest cellular uptake by Ana-1 cells in plasma-suspended RPMI 1640 media (Figure 2k). In experimental studies on nanoparticles, a standard practice that aims to extend a nanoparticle's blood circulation lifetime is to decrease the particle's *in vitro* cellular uptake especially by phagocytes like macrophages, assuming that lower *in vitro* cellular uptake may correspond to lower uptake *in vivo* by the reticuloendothelial system (RES), which is a heterogeneous population of phagocytic cells in systemically fixed tissues that play an important role in the clearance of particles and soluble substances in the circulation and tissues. But, a particle's clearance from blood is so complex that cellular uptake presents just one aspect that contributes to the particle's clearance from circulation, and other processes involved include but are not limited to filtering by organs such as kidney (Supplementary Table 12) (*Nat. Biotechnol.*, **2015**, 33, 941; *Angew. Chem., Int. Ed.*, **2014**, 53, 12320). In fact, that uptake of a nanoparticle by blood cells like phagocytes is not the sole factor that determines the nanoparticle's blood circulation lifetime (*Angew. Chem. Int. Ed.*, **2014**, 53, 12320) may explain why 75kPa@lipid and 100kPa@lipid (our second- and third-softest nanoparticles, respectively) exhibited comparably long blood clearance half-lives (Figure 2c) despite of their distinct cellular uptake efficiencies by suspended Ana-1 cells (Figure 2k) — where 75kPa@lipid exhibited the second-highest cellular uptake whereas 100kPa@lipid the lowest.

Our plot on the relationship between nanoparticle blood circulation lifetime *versus* nanoparticle elasticity (Figure 2c) revealed that the relationship is non-monotonic, rather than monotonic (*i.e.*, longer systemic circulation for softer nanoparticles) as many prior reports on this topic have claimed (Supplementary Table 1). Of note, this non-monotonic relationship between nanoparticle blood clearance lifetime *versus* nanoparticle elasticity we observed was reached based on a compiled plot that included not only our model nanoparticles but also nanoparticles in previous reports on this topic (Figure 2c). In fact, when carrying out the study, what stimulated us to include particles from prior reports into our plot on the relationship between nanoparticle blood circulation lifetime *versus* nanoparticle elasticity was the observation that our liposome (DOPC: DSPE-PEG₂₀₀₀ = 90:10), the softest member of our model nanoparticle family, surprisingly exhibited the shortest blood clearance half-life (Figure 2c).

Encouraged by this and to continue clarifying our own confusion then (*i.e.*, the softest model nanoparticle exhibiting the shortest blood clearance half-life), we next included the particles from prior reports into our plot of on the relationship between nanoparticle blood circulation lifetime *versus* nanoparticle elasticity (Figure 2c) and hoped to get a bird's eye view on the whole situation. To our own surprise, we found a non-monotonic relationship of nanoparticle blood circulation lifetime *versus* nanoparticle elasticity

(Figure 2c) that can actually be divided into three distinct regions depending on nanoparticle elasticity, which are Region I where nanoparticle elasticity is < 15 kPa, Region II where nanoparticle elasticity is 15-75 kPa, and Region III where nanoparticle elasticity is >75 kPa. Within each individual one of these three regions we observed, a nanoparticle of lower elasticity generally tends to exhibit longer blood clearance lifetime, a trend consistent with what has been claimed in prior reports on this topic (*ACS Nano*, **2015**, 9, 11628; *ACS Nano*, **2015**, 9, 3169; *ACS Nano*, **2012**, 6, 6681; *Proc. Natl. Acad. Sci. U. S. A.*, **2011**, 108, 586). When comparing nanoparticles from differing regions, we found that those from Region I (*i.e.*, the softest particle) generally tend to exhibit longer blood clearance half-lives than those from either Region II or Region III, which again supports the previously claimed trend that softer nanoparticles have longer blood circulation lifetimes (Supplementary Table 1). What's surprisingly is that, compared with particles from Region I and Region III, those from Region II — which included not only liposomes but also very soft hydrogel particles (Table R1) — exhibited the shortest blood clearance half-lives (Figure 2c). From this perspective, we have to admit that, if we did not include the liposome as our softest nanoparticle at the very beginning, we may have missed the chance to find the non-monotonic nature of the relationship of nanoparticle blood circulation lifetime *versus* nanoparticle elasticity (Figure 2c), not even to mention the three aforementioned distinct regions. Once particles from prior reports on this topic have been included into our plot of blood clearance half-life *versus* nanoparticle elasticity (Figure 2c), the importance of our liposome (DOPC: DSPE-PEG₂₀₀₀ = 90:10) became reduced simply because it is not the sole particle that falls into Region II anymore; in fact, even if we completely remove it from this plot (Figure 2c), the existence of Region II in this plot and the non-monotonic nature for the relationship between blood clearance half-life *versus* nanoparticle elasticity will still retain. But, again, we still have to admit that, without this liposome, we may not get the chance of reaching our current conclusion.

Table R1. Nanoparticles covered in Region II of Figure 2c.

Nanoparticle	Composition	Ref.
Red Blood Cell Mimics Hydrogel (5.2–5.9 μm in diameter and 1.22–1.54 μm tall)	HEA ^a /PEGDA ^b	Proc. Natl. Acad. Sci. U. S. A. , 2011 , 108, 586
Discoidal Polymeric Nanoconstructs (~1000 nm in diameter and ~400 nm tall)	PLGA ^c /PEGDA/lipid-DOTA	ACS Nano , 2015 , 9, 11628
liposomes	HSPC/Chol (3:1) HSPC/Chol/DSPE-mPEG-2000 (3:1:1)	J. Controlled Release , 2007 , 120, 161
liposomes	PC/Chol/PEG-PE (1:1:0.16)	FEBS Lett. , 1990 , 268, 235
liposomes	SPC/Chol/PEGylated UA ^d (50:8:5)	J. Nanopart. Res. , 2016 , 18, 34
liposomes	DOPE/CHEMS ^e (1.5:1)	Drug Delivery Transl. Res. , 2019 , 9, 123
liposomes	HSPC/Chol/DSPE-PEG-2000 (57:38:5)	Nanoscale , 2020 , 12, 18875
liposomes	HSPC/Chol/PEG	Mol. Pharmaceutics , 2020 , 17, 472

^a HEA denotes 2-hydroxyethyl acrylate; ^b PEGDA denotes poly(ethylene glycol) diacrylate; ^c PLGA denotes poly(D,L-lactide-co-glycolide); ^d UA denotes ursolic acid; ^e CHEMS denotes cholesteryl hemisuccinate.

Two reasons we think may underlie why prior studies on the relationship between blood circulation lifetime *versus* nanoparticle elasticity have missed the observation of the three aforementioned regions but instead claimed a monotonic relationship between blood circulation lifetime *versus* particle elasticity (Supplementary Table 1), which are:

(1) these prior studies compared the performances of their own particles but forgot to include particles from previous reports on this topic or other relevant topics, and (2) these prior studies unconsciously used particles from a same Region or compared particles from Region I with those from Region II or III (as summarized in Supplementary Table 10). To our best knowledge, we did not find any prior report that compared particles from Region II with those from Region III or included particles across the three aforementioned regions. Once particles from Region II were included in the comparison, prior conclusion that softer particles have longer blood circulation lifetimes became partially right, applicable only when we are not comparing particles from Region II with those from Region III. Fortunately, our model nanoparticle family has its softest member, the liposome composed of DOPC: DSPE-PEG₂₀₀₀ = 90:10, that belongs to Region II and our plot of nanoparticle blood circulation lifetime *versus* nanoparticle elasticity (Figure 2c) included particles from prior reports on similar topics rather than just our own model nanoparticles, both of which contributed crucially to our eventual observation of the three aforementioned regions and consequently the non-monotonic relationship between nanoparticle blood circulation lifetime *versus* nanoparticle elasticity. From this perspective, the liposome is an indispensable member from our model nanoparticle family.

Supplementary Table 10. Summary of elasticity region of the particles chose to study the relationship of elasticity and blood circulation in literature.

Elasticity	Nanoparticle	Ref.
Region I	discoidal polymeric nanoconstructs	ACS Nano , 2015 , 9, 11628
Region III	zwitterionic nanogels	ACS Nano , 2012 , 6, 6681
	silica nanocapsules	ACS Nano , 2018 , 12, 2846
Regions I and II	hydrogel microparticles	Proc. Natl. Acad. Sci. U. S. A. , 2011 , 108, 586
Regions I and III	PEG-based hydrogel nanoparticles	ACS Nano , 2015 , 9, 3169

Beside stimulating us to include particles from prior reports on related topic into our plot of blood circulation lifetime *versus* particle elasticity (Figure 2c), our liposome DOPC: DSPE-PEG₂₀₀₀ = 90:10 was very important in our assays for unveiling the effects of the relative ApoA1 abundance in corona on nanoparticle blood circulation lifetime (Figure 4), thanks to the liposome's nearly 0% relative ApoA1 abundance in corona of (Figure 3f and Supplementary Figure 13a) and hence its use as a valuable negative control (Figure 4).

(4) *The conclusion about the correlation between the relative abundance apolipoprotein A-I in corona and their circulation time is bold, as shown in Fig. 3e, the adsorption of other proteins such as complement proteins varies a lot with the particle elasticity. So the analysis of the correlation between the protein corona and circulation time is too simplified, no clear trend doesn't mean no contribution.*

Author Response: We thank the reviewer for the critical comments and inspiring suggestion. In our previous manuscript, our calculations on the correlation coefficient (*i.e.*, Pearson's r in this work) between the relative protein abundance in corona and our nanoparticles blood clearance half-lives (Figure 3h) show that, although each protein in corona may contribute positively or negatively to the particle blood circulation lifetime and has its corresponding calculated correlation coefficient with a value large or small, ApoA1 is the only one that exhibits a Pearson's r of >0.7 , indicative of strong positive correlation, suggesting ApoA1 in corona as the key player in the blood circulation lifetime for our model nanoparticles. In revision, to experimentally examine whether the relative abundance of ApoA1 in corona has significant positive correlation with nanoparticle blood circulation lifetime as our calculations on correlation coefficient (*i.e.*, Pearson's r in this work) (Figure 3h) show in the previous manuscript, we examined whether screening/shielding the adsorbed ApoA1 in corona leads to shortened blood circulation lifetime for nanoparticles over which ApoA1 enjoys high relative abundance in corona. So we used ApoA1 antibody to achieve the goal of shielding the adsorbed ApoA1 in corona, the ApoA1 antibody located the outer layer of the ApoA1 protein and prevented the interaction between ApoA1 and cells due to the specific recognition and combination between ApoA1 and its antibody (*Proc. Natl. Acad. Sci. U. S. A.*, **2021**, 118, e2108131118). To this end, we carried out similar blood circulation assays as those performed for obtaining the plot on the relationship between nanoparticle blood circulation lifetime *versus* nanoparticle elasticity (Figure 2c) but added an extra step — which is the pre-treatment with ApoA1 antibody — in between the incubation of a particle (for 12h) with mouse plasma and the intravenous injection of the particle into healthy mouse for blood retention tests (Figure 3i); specifically, we incubated a model nanoparticle (for 12h) with mouse plasma to obtain nanoparticle-protein complex, treated (for 1h) the resulting nanoparticle-complex with ApoA1 antibody (at a molar ratio of ApoA1 antibody to estimated ApoA1 amount in corona $\sim 1:1$) in effort to screen/shield the nanoparticle surface-adsorbed ApoA1 with ApoA1 antibody, and then injected the as-treated particle through tail vein into healthy mouse for blood retention tests to examine how screening/shielding ApoA1 in corona of a nanoparticle affects the particle's blood circulation lifetime (Figure 3i). In these additional blood circulation assays, liposome (DOPC: DSPE-PEG₂₀₀₀ = 90:10), 188kPa@lipid, and PLGA@lipid were used as the representatives for nanoparticles with extremely low ApoA1 abundance in corona, nanoparticles with appreciably high ApoA1 abundance in corona, and nanoparticles with intermediate ApoA1 abundance in corona, respectively; ApoA1 abundance in corona is observed to be $\sim 0\%$ for liposome, $\sim 8\%$ for PLGA@lipid, and $\sim 26\%$ for 188kPa@lipid (Figure 3f and Supplementary Figure 13a). Our addition blood circulation assays (Figure 3j) show that,

for liposome (DOPC: DSPE-PEG₂₀₀₀ = 90:10), pre-treating the liposome-protein complex with ApoA1 antibody failed to appreciably affect the complex's blood clearance half-life (9 h and 9.6 h with and without the ApoA1 antibody treatment, respectively) (Figure 3j), which is easy to understand considering the extremely low abundance of ApoA1 in its corona (Figure 3f and Supplementary Figure 13a). In stark contrast, for 188kPa@lipid nanoparticle, pre-treating the nanoparticle-protein complex with ApoA1 antibody significantly reduced the complex's blood clearance half-life (Figure 3j), with the calculated average blood clearance half-life being shortened from 44 h to 36 h (relatively by ~18%) due to the shielding of adsorbed ApoA1 by ApoA1 antibody. For PLGA@lipid, the pre-treatment with ApoA1 antibody reduced the complex's blood clearance half-life (with the calculated average blood clearance half-life being shortened from 39 h to 35 h) (relatively by ~10%) (Figure 3j). Interestingly, for these three model nanoparticles, their order of relative change in blood clearance (0% for liposome, 10% for PLGA@lipid, and ~18% for 188kPa@lipid) nicely mirrors their ranks of ApoA1 abundance in corona (~0% for liposome, ~8% for PLGA@lipid, and ~26% for 188kPa@lipid) (Figure 3f and Supplementary Figure 13a). Clearly, shielding/screening adsorbed ApoA1 in corona over a nanoparticle shortens the particle's blood circulation lifetime and the relative extent of such shortening is related with the particle's relative abundance of ApoA1 in corona. Combined with our calculations on the correlation coefficient between relative protein abundance in corona *versus* nanoparticle blood circulation lifetime (Figure 3h), these additional blood circulation assays demonstrate that ApoA1 abundance in corona is correlated significantly and positively with nanoparticle circulation lifetime in blood.

Figure 3. (f) Heat map of the most abundant proteins in protein coronas of our nanoparticles. (g) Distribution of the relative abundance in corona on different particles and (h) Pearson's r between the relative abundance in corona and nanoparticle blood clearance half-life for proteins which exhibited a relative abundance of $>5\%$ on at least one nanoparticle. Pearson's r of >0.6 and <-0.6 indicate strong positive and negative correlations, respectively. (i) Schematic illustration on blood circulation test of our nanoparticles with screening/shielding the adsorbed ApoA1 in corona by ApoA1 antibody. (j) Blood clearance half-lives of our nanoparticles with screening/shielding the adsorbed ApoA1 in corona by ApoA1 antibody.

In revision, we also calculated the correlation coefficient (*i.e.*, Pearson's r in this work) between the relative abundance of physiologically functional proteins in corona and our nanoparticles' blood clearance half-lives (Figure R2 below) as the reviewer suggested and found that only immunoglobulins and coagulation proteins consistently exhibited a calculated Pearson's r of <-0.6 in both trials, indicative of strong negative correlation. Notably, no protein family consistently exhibited a calculated Pearson's r of >0.6 in both trials, indicative of lack of strong positive correlation.

corona protein according to physiological functions *versus* blood clearance half-life

Figure R2. Pearson's r between the relative abundance of proteins in corona which were classified according to physiological functions and nanoparticle blood clearance half-life.

A previous paper found complement activation could not explain differences in the clearance of nanoparticles in animals deficient in the C3 protein and adsorption of apolipoproteins prolongs nanoparticle circulation lifetime (*Nat. Commun.*, **2017**, 8, 777). This indicated that the complement proteins probably didn't work in regulation of a nanoparticle's circulation time.

(5) *The fundamental understanding of the role of apolipoprotein A-I in protein corona and consequently in controlling circulation time is lacking. There are many studies about protein corona and the role of apolipoprotein A-I in controlling NP biological identities. More in depth discussion should be included.*

Author Response: We thank the reviewer for the critical comments and inspiring suggestion. In the previous manuscript, we found that, among the most abundant proteins in coronas over our nanoparticles, ApoA1 is the only one whose relative abundance in corona correlates significantly (Pearson's $r > 0.6$) with the blood clearance lifetime for our nanoparticles (Figure 3h). As the correlation between ApoA1 abundance in corona *versus* nanoparticle blood circulation lifetime is both significant and positive, we in revision moved one step further to examine whether screening/shielding the adsorbed ApoA1 in corona leads to shortened blood circulation lifetime for nanoparticles over which ApoA1 enjoys high relative abundance in corona. So we used ApoA1 antibody to achieve the goal of shielding the adsorbed ApoA1 in corona, the ApoA1 antibody located the outer layer of the ApoA1 protein and prevented the interaction between ApoA1 and cells due to the specific recognition and combination between ApoA1 and its antibody (*Proc. Natl. Acad. Sci. U. S. A.*, **2021**, 118, e2108131118). To this end, we carried out similar blood circulation assays as those performed for obtaining the plot on the relationship between nanoparticle blood circulation lifetime *versus* nanoparticle elasticity (Figure 2c) but added an extra step — which is the pre-treatment with ApoA1 antibody — in between the incubation of a particle (for 12 h) with mouse plasma and the intravenous injection of the particle into

healthy mouse for blood retention tests (Figure 3i); specifically, we incubated a model nanoparticle (for 12 h) with mouse plasma to obtain nanoparticle-protein complex, treated (for 1h) the resulting nanoparticle-complex with ApoA1 antibody (at a molar ratio of ApoA1 antibody to estimated ApoA1 amount in corona ~ 1:1) in effort to screen/shield the nanoparticle surface-adsorbed ApoA1 with ApoA1 antibody, and then injected the as-treated particle through tail vein into healthy mouse for blood retention tests to examine how screening/shielding ApoA1 in corona of a nanoparticle affects the particle's blood circulation lifetime (Figure 3i). In these additional blood circulation assays, liposome (DOPC: DSPE-PEG₂₀₀₀ = 90:10), 188kPa@lipid, and PLGA@lipid were used as the representatives for nanoparticles with extremely low ApoA1 abundance in corona, nanoparticles with appreciably high ApoA1 abundance in corona, and nanoparticles with intermediate ApoA1 abundance in corona, respectively; ApoA1 abundance in corona is observed to be ~0% for liposome, ~8% for PLGA@lipid, and ~26% for 188kPa@lipid (Figure 3f and Supplementary Figure 13a). Our addition blood circulation assays (Figure 3j) show that, for liposome (DOPC: DSPE-PEG₂₀₀₀ = 90:10), pre-treating the liposome-protein complex with ApoA1 antibody failed to appreciably affect the complex's blood clearance half-life (9 h and 9.6 h with and without the ApoA1 antibody treatment, respectively) (Figure 3j), which is easy to understand considering the extremely low abundance of ApoA1 in its corona (Figure 3f and Supplementary Figure 13a). In stark contrast, for 188kPa@lipid nanoparticle, pre-treating the nanoparticle-protein complex with ApoA1 antibody significantly reduced the complex's blood clearance half-life (Figure 3j), with the calculated average blood clearance half-life being shortened from 44 h to 36 h (relatively by ~18%) due to the shielding of adsorbed ApoA1 by ApoA1 antibody. For PLGA@lipid, the pre-treatment with ApoA1 antibody reduced the complex's blood clearance half-life (with the calculated average blood clearance half-life being shortened from 39 h to 35 h) (relatively by ~10%) (3j). Interestingly, for these three model nanoparticles, their order of relative change in blood clearance (0% for liposome, 10% for PLGA@lipid, and ~18% for 188kPa@lipid) nicely mirrors their ranks of ApoA1 abundance in corona (~0% for liposome, ~8% for PLGA@lipid, and ~26% for 188kPa@lipid) (Figure 3f and Supplementary Figure 13a). Clearly, shielding/screening adsorbed ApoA1 in corona over a nanoparticle shortens the particle's blood circulation lifetime and the relative extent of such shortening is related with the particle's relative abundance of ApoA1 in corona, which confirms the significantly positive correlation between the relative ApoA1 abundance in corona *versus* nanoparticle blood circulation lifetime (Figure 3h).

The reasons why ApoA1 abundance in corona is correlated positively with nanoparticle blood circulation lifetime may lie in the following aspects: (1) ApoA1 preferentially interacts with nanoparticles of intermediate elasticity over those of extremely soft or stiff elasticity, as evidenced by the isothermal titration calorimetry assays (Figure 4a) using liposome (DOPC: DSPE-PEG₂₀₀₀ = 90: 10), 188kPa@lipid, and PLGA@lipid as representatives for particles of extremely soft, intermediate, and stiff elasticity, respectively. (2) ApoA1 is an effective dysopsonic protein; in fact, as dysopsonic protein, ApoA1 is more efficient than bovine serum albumin (BSA) — a widely used

model dysopsonic protein (*ACS Nano*, **2013**, 7, 10960) (Figure 4b), as demonstrated by the observations that pre-treating a nanoparticle with ApoA1 significantly reduced the particle's subsequent uptake by Ana-1 macrophage cells no matter whether the cells are adherent or suspended in cell culture whereas similar pre-treatment with BSA reduced the particle's subsequent uptake only by adherent Ana-1 cells and to a slight extent (Figure 4c-d).

In revision, we further collected references that have reported the detection of ApoA1 in corona (Supplementary Table 11 and Table R2) and summarized the reported roles of ApoA1 in corona in these references (Supplementary Table 11). Consistent with our findings in this work, some references show that roles of ApoA1 in corona include (1) reducing the nanoparticle's cellular uptake (*Proc. Natl. Acad. Sci. U. S. A.*, **2021**, 118, e2108131118; *Colloids Surf., B*, **2019**, 173, 891; *Nanoscale*, **2019**, 11, 10727) and (2) extending the particle's circulation lifetime in blood (*Colloids Surf., B*, **2019**, 173, 891). Since these prior studies used stiff nanoparticles as the models (*Proc. Natl. Acad. Sci. U. S. A.*, **2021**, 118, e2108131118; *Colloids Surf., B*, **2019**, 173, 891; *Nanoscale*, **2019**, 11, 10727; *Nano Lett.*, **2019**, 19, 1260), they did not find the preferentiality of ApoA1 in its interaction with nanoparticles we observed in this work.

Supplementary Table 11. The previous works in which ApoA1 proteins appeared in the protein corona of particles.

Nanoparticle	Ranking of ApoA1	Roles of ApoA1 in protein corona	Ref.
Silica nanoparticles	1	Decrease cellular uptake, relieve cytotoxicity and proinflammatory effect	Proc. Natl. Acad. Sci. U. S. A. , 2021 , 118, e2108131118
PEGylated graphene oxide coated gold nanorods	1	Decrease macrophage uptake and prolong circulation lifetime	Colloids Surf., B , 2019 , 173, 891
Polystyrene nanoparticles	N.D.	Decrease cellular uptake	Nanoscale , 2019 , 11, 10727
Graphene nanoflakes	N.D.	Mediate cellular uptake through recognition of SR-B1 receptor	Nano Lett. , 2019 , 19, 1260
75-700kPa@lipid	1	Function as a dysopsonic protein and prolong circulation lifetime	This work

*N.D. denotes 'not defined'.

Table R2. The previous works in which apolipoproteins appeared in the protein corona of particles.

Nanoparticle	Ranking of ApoA1	Roles of apolipoproteins in protein corona	Ref.
PEG-Polystyrene	2	Contribute to the stealth effect	Nat. Nanotechnol. , 2016 , 11, 372
PEGylated doxorubicin-encapsulated liposomes	> 10	Favor long blood circulation	Adv. Mater. , 2019 , 31, 1803335
Lipid solid nanoparticles	N.D.	Affect cellular uptake (apolipoprotein E)	Nanoscale , 2019 , 11, 8760
Multi-component (MC) liposome-PEG ₁₀₀₀	1	Bind selectively with scavenger receptor class B type 1 (SR-BI)	Nanoscale , 2014 , 6, 2782
Multi-component (MC) liposome-PEG ₂₀₀₀	> 10		
PEGylated liposomes	6	N.D.	Biochim. Biophys. Acta, Biomembr. , 2016 , 1858, 189
5% PEG-thermosensitive liposome	7	N.D.	J. Controlled Release , 2021 , 333, 1
10% PEG-thermosensitive liposome	6		
Carbohydrate-Based Nanocarriers	4	N.D.	Angew. Chem. Int. Ed. , 2015 , 54, 7436
Graphene nanoflakes	2	N.D.	Nat. Commun. , 2018 , 9, 1577
PEG-PLGA	7	N.D.	J. Controlled Release , 2018 , 287, 121
Nano-graphene oxide (nGO)-PEG	6	N.D.	Nanoscale , 2018 , 10, 10863

*N.D. denotes 'not defined'.

Response to Reviewer #2 (Remarks to the Author):

In their manuscript entitled "Nanoparticle Elasticity Affects Systemic Circulation by Modulating the Adsorption of Apolipoprotein A-I in Corona Formation" the authors investigate the effects of nanoparticle core density on the protein corona and systemic circulation.

The study seems overall well designed and results are clearly presented. The study could be an important contribution to the field, as it could provide a basis for the yet unknown mechanisms why nanoparticles of different elasticity show different biological behaviours (e.g. uptake, circulation etc).

However, I have one major point that requires to be addressed in a revision:

A key aspect of the study is the investigation of the protein corona and its characterization by LC-MS based proteomics.

Unfortunately, the proteomics data are very insufficiently described. No data are provided except on a highly aggregated level (i.e. figures).

It is not even clear if the experiments or analyses were performed in replicates, or if data from an n=1 proteomic analysis are shown. As for all experiments, multiple biological replicates are required. I also highly recommend the analysis of multiple technical replicates to achieve reliable quantification values using a spectral counting-based quantification approach.

Unfortunately, the authors also chose not to present any quantification data, making it completely impossible to judge the validity of proteomic results.

The authors must provide the entire proteomics datasets (identification and quantification results) in the supplementary information. Data should include (at least): Protein Identifier, Scores(peptide and protein level), FDR, number of peptides per proteins, quantification values from all biological and technical replicates.

The methods for proteomic analysis are very insufficiently described.

- No details at all regarding LC conditions (e.g. instrument, column, solvents, flow rates, gradients, temperature, injection amount per replicate....) are provided.

- No details at all on instrument settings or acquisition settings are provided.

- Not even the most basic details for database search parameters (name and version of database search engine, name and version of database, mass tolerances, FDR, modifications, etc) are provided.

Please refer to MIAPE guidelines for a proper reporting of proteomics datasets.

Additionally, the proteomic datasets (rawdata and search results) need to be submitted to a public repository (i.e. ProteomeXchange) to allow other researchers to access the data.

Author Response: We thank the reviewer for the critical comments and inspiring suggestion. In the revision, we added more details of the LC-MS based proteomics in the **“17. Liquid-chromatography mass-spectrometry (LC-MS) analysis in MATERIALS AND METHODS”** section of supplementary information and the proteomics data which contains the protein identifier, scores (peptide and protein level), FDR, number of peptides per proteins and other information have been provided in our file of source data excel.

Briefly, the dissolved peptide sample was then analyzed by high-performance liquid chromatography (HPLC) system (Thermo scientific, model EASY-nLC 1200 system) coupled with a mass spectrometer (Thermo scientific, model Q Exactive Plus). Tryptic peptides were separated on the EASY-nLC 1200 system equipped with a C18 analytical

reversed-phase column (particle size: 2 μm ; pore size: 100 \AA ; diameter \times length: 50 μm \times 150 mm; Acclaim[®] PepMap[™] RSLC, thermo scientific) and a C18 trap column (particle size: 5 μm ; pore size: 100 \AA ; diameter \times length: 100 μm \times 20 mm; Acclaim[®] PepMap[™] 100, thermo scientific). The samples were processed with mobile phase solvent A consisting of 0.1% (v/v) formic acid in pure water, and mobile phase solvent B was 80% acetonitrile with 0.1% (v/v) formic acid in water. The separation was performed at a sample flow rate of 0.3 $\mu\text{l}/\text{min}$, using a gradient of 3-35% solvent B over 100 min. Typical sample injection volume was 1 μL .

The high performance liquid chromatography (HPLC) system was on-line coupled with a mass spectrometer (Thermo scientific, model Q Exactive Plus), and the mass spectrometer was set at the data-dependent mode to acquire MS/MS data. Electrospray ionization (ESI) was performed in positive ion mode. The ionization voltage was 2 kV. The capillary temperature was set at 320 $^{\circ}\text{C}$. The normalized collision energy was at 27%, and the default charge state was at 2. Data were acquired within a range of m/z 150-2000 Da in one full scan.

The mass spectrometric data then were used to search against the UniProt protein database with Thermo Proteome Discoverer software suite (version: 2.2.0.388). During database searches, the protein strict and relaxed false discovery rates were set at 1% and 5%, respectively. The mass tolerances for precursor mass and fragment mass were 10 ppm and 0.02 Da, respectively. The following criteria were used for the search: one missed cleavage, fixed carbamidomethyl modification for cysteine, and variable oxidation for methionine. And the relative abundance of a specific protein in the corona of a nanoparticle was determined through the method of spectral counting (SpC).

We completely independently performed this trail for twice on each sample, the results which contains the protein identifier, scores (peptide and protein level) and other information have been provided in our source data excel. And all the raw proteomic datasets have been submitted to a public repository iProX to allow all other researchers to access the data. The mass spectrometry proteomics data have been deposited to the ProteomeXchange Consortium (<http://proteomecentral.proteomexchange.org>) via the iProX partner repository with the dataset identifier PXD034004.

REVIEWERS' COMMENTS

Reviewer #1 (Remarks to the Author):

The paper has been improved after addressing the questions raised by the reviewers, but a general comment is that what has been observed for this particular system might not be able to apply to other systems, so it is not a general rule or principle rather a system specific phenomenon, as it is well accepted that soft nanoparticles tend to circulate longer (e.g. the first cancer nanomedicine, Doxil, liposomal doxorubicin, which has very long circulation time), and many other studies have also demonstrated it (Softer Zwitterionic Nanogels for Longer Circulation and Lower Splenic Accumulation. ACS Nano 2012, 6, 6681–6686; Elasticity of Nanoparticles Influences Their Blood Circulation, Phagocytosis, Endocytosis, and Targeting. ACS Nano 2015, 9, 3169–3177).

In terms of the surface charge, the authors referred to the basics as well as many other researchers, explained how the core could affect the zeta potential. Very similarly, Moses et al. (NATURE COMMUNICATIONS, (2018) 9:130) reported nanolipogels (NLGs) composed of identical lipid bilayers encapsulating an alginate core, with tunable elasticity, which is similar to the systems proposed in this work. The NLG particles with DOPC as the lipid bilayer have very similar zeta potential (about -6 to -9 mV), which is the range we could consider similar. In contrast, this work used DOPC + DSPE-PEG2000 as the lipid bilayer which have such negative and different zeta potential (-15.1 ~ -28.7 mV) compared to Moses's despite it is known that DSPE-PEG2000 is to screen the charge. It is known that the charge of nanoparticles affect the protein corona significantly. The blood clearance seems have some correlation with the charge, the more negative surface charge the quicker clearance rate. Also, the big variance (big error bar) of the amounts of adsorbed proteins (Figure 3a) make the quantitative comparison less meaningful.

The claim "no report to our best knowledge has systematically examined the effects of nanoparticle elasticity on protein corona" is not correct. Tengjisi et al. (Influence of nanoparticle mechanical property on protein corona formation, JCIS, 2022, 606, 1737-1744). They found the protein corona of the stiffest nanocapsules contained the highest amount of complement protein (Complement C3) and immunoglobulin proteins, which contributed to their high macrophage uptake.

English needs to be significantly improved. For example, the first two sentences of the abstract, "Nanoparticle elasticity is crucial in nanoparticles' physiological fate. How nanoparticle elasticity does so remains, however, unknown", " a same PEGylated".

Reviewer #2 (Remarks to the Author):

The authors have sufficiently addressed my comments. I recommend publication of this manuscript.

Reviewer #1 (Remarks to the Author):

The paper has been improved after addressing the questions raised by the reviewers, but a general comment is that what has been observed for this particular system might not be able to apply to other systems, so it is not a general rule or principle rather a system specific phenomenon, as it is well accepted that soft nanoparticles tend to circulate longer (e.g. the first cancer nanomedicine, Doxil, liposomal doxorubicin, which has very long circulation time), and many other studies have also demonstrated it (Softer Zwitterionic Nanogels for Longer Circulation and Lower Splenic Accumulation. ACS Nano 2012, 6, 6681–6686; Elasticity of Nanoparticles Influences Their Blood Circulation, Phagocytosis, Endocytosis, and Targeting. ACS Nano 2015, 9, 3169–3177).

Author Response: We thank the reviewer for the critical comments. But, the non-monotonic relationship between nanoparticle blood clearance lifetime *versus* nanoparticle elasticity (Figure 2c) we observed should be a general rule, rather than a system specific phenomenon applicable exclusively to our model nanoparticle system, for reasons as follow.

Firstly, this non-monotonic relationship between nanoparticle blood clearance lifetime *versus* nanoparticle elasticity (Figure 2c) we observed was reached based on a compiled plot that included not only our model nanoparticles but also nanoparticles in previous reports on this topic. In fact, this compiled plot (Figure 2c) has already included the two studies mentioned here by the reviewer (“Softer Zwitterionic Nanogels for Longer Circulation and Lower Splenic Accumulation” (ACS Nano, 2012, 6, 6681–6686) and “Elasticity of Nanoparticles Influences Their Blood Circulation, Phagocytosis, Endocytosis, and Targeting” (ACS Nano, 2015, 9, 3169–3177)).

Secondly, this relationship (Figure 2c) can be divided into three distinct regions depending on nanoparticle elasticity, which are Region I where nanoparticle elasticity is < 15 kPa, Region II where nanoparticle elasticity is 15-75 kPa, and Region III where nanoparticle elasticity is >75 kPa. (1) Within each individual one of these three regions we observed, a nanoparticle of lower elasticity generally tends to exhibit longer blood clearance lifetime, a trend consistent with what has been claimed in prior reports on this topic (ACS Nano, 2015, 9, 11628; ACS Nano, 2015, 9, 3169; ACS Nano, 2012, 6, 6681; Proc. Natl. Acad. Sci. U. S. A., 2011, 108, 586). For example, the study titled “Softer Zwitterionic Nanogels for Longer Circulation and Lower Splenic Accumulation” (ACS Nano, 2012, 6, 6681–6686) compared the blood circulation lifetimes of nanoparticles whose elasticity values (modulus of bulk hydrogels: 0.18-1.35 MPa) fall into a same region (*i.e.*, Region III) and observed longer circulation for softer nanoparticles. (2) When comparing nanoparticles from different regions, we found that those from Region I (*i.e.*, the softest particle) generally tended to exhibit longer blood clearance half-lives than those from either Region II or Region III, which again supports the previously claimed trend that softer nanoparticles have longer blood circulation lifetimes (Supplementary Table 1). For example, the study titled “Elasticity of Nanoparticles

Influences Their Blood Circulation, Phagocytosis, Endocytosis, and Targeting” (*ACS Nano*, **2015**, 9, 3169–3177) compared the blood circulation lifetimes of two nanoparticles differing in elasticity and observed that the soft nanoparticle which has an elasticity of 10 kPa and therefore belongs to Region I exhibited longer circulation lifetime in blood than the hard nanoparticle which has an elasticity of 3,000 kPa and therefore belongs to Region III. (3) What’s surprisingly is that, compared with particles from Region I and Region III, those from Region II - which included not only liposomes but also very soft hydrogel particles (Supplementary Table 13) - exhibited the shortest blood clearance half-lives (Figure 2c).

Supplementary Table 13. Nanoparticles covered in Region II of Figure 2c.

Nanoparticle	Composition	Ref.
Red Blood Cell Mimics Hydrogel (5.2–5.9 μm in diameter and 1.22–1.54 μm tall)	HEA ^a /PEGDA ^b	Proc. Natl. Acad. Sci. U. S. A. , 2011 , 108, 586
Discoidal Polymeric Nanoconstructs (~1000 nm in diameter and ~400 nm tall)	PLGA ^c /PEGDA/lipid-DOTA	ACS Nano , 2015 , 9, 11628
liposomes	HSPC/Chol (3:1) HSPC/Chol/DSPE-mPEG-2000 (3:1:1)	J. Controlled Release , 2007 , 120, 161
liposomes	PC/Chol/PEG-PE (1:1:0.16)	FEBS Lett. , 1990 , 268, 235
liposomes	SPC/Chol/PEGylated UA ^d (50:8:5)	J. Nanopart. Res. , 2016 , 18, 34
liposomes	DOPE/CHEMS ^e (1.5:1)	Drug Delivery Transl. Res. , 2019 , 9, 123
liposomes	HSPC/Chol/DSPE-PEG-2000 (57:38:5)	Nanoscale , 2020 , 12, 18875
liposomes	HSPC/Chol/PEG	Mol. Pharmaceutics , 2020 , 17, 472

^a HEA denotes 2-hydroxyethyl acrylate; ^b PEGDA denotes poly(ethylene glycol) diacrylate; ^c PLGA denotes poly(D,L-lactide-co-glycolide); ^d UA denotes ursolic acid; ^e CHEMS denotes cholesteryl hemisuccinate.

Two reasons we think may underlie why prior studies on the relationship between blood circulation lifetime *versus* nanoparticle elasticity have missed the observation of these three aforementioned regions but instead claimed a monotonic relationship between blood circulation lifetime *versus* particle elasticity (Supplementary Table 1), which are: (1) these prior studies compared the performances of their own particles but forgot to

include particles from previous reports on this topic or other relevant topics, and (2) these prior studies unconsciously used particles from a same Region or compared particles from Region I with those from Region II or III (as summarized in Supplementary Table 10). To our best knowledge, we did not find any prior report that compared particles from Region II with those from Region III or included particles across the three aforementioned regions. Once particles from Region II were included in the comparison, prior conclusion that softer particles have longer blood circulation lifetimes became partially right, applicable only when we are not comparing particles from Region II with those from Region III. Fortunately, our model nanoparticle family has its softest member, the liposome composed of DOPC: DSPE-PEG₂₀₀₀ = 90:10, that belongs to Region II and our plot of nanoparticle blood circulation lifetime *versus* nanoparticle elasticity (Figure 2c) included particles from prior reports on similar topics rather than just our own model nanoparticles, both of which contributed crucially to our eventual observation of the three aforementioned regions and consequently the non-monotonic relationship between nanoparticle blood circulation lifetime *versus* nanoparticle elasticity.

Supplementary Table 10. Summary of elasticity region of the particles chose to study the relationship of elasticity and blood circulation in literature.

Elasticity	Nanoparticle	Ref.
Region I	discoidal polymeric nanoconstructs	ACS Nano , 2015 , 9, 11628
Region III	zwitterionic nanogels	ACS Nano , 2012 , 6, 6681
	silica nanocapsules	ACS Nano , 2018 , 12, 2846
Regions I and II	hydrogel microparticles	Proc. Natl. Acad. Sci. U. S. A. , 2011 , 108, 586
Regions I and III	PEG-based hydrogel nanoparticles	ACS Nano , 2015 , 9, 3169

In terms of the surface charge, the authors referred to the basics as well as many other researchers, explained how the core could affect the zeta potential. Very similarly, Moses et al. (NATURE COMMUNICATIONS, (2018) 9:130) reported nanolipogels (NLGs) composed of identical lipid bilayers encapsulating an alginate core, with tunable elasticity, which is similar to the systems proposed in this work. The NLG particles with DOPC as the lipid bilayer have very similar zeta potential (about -6 to -9 mV), which is the range we could consider similar. In contrast, this work used DOPC + DSPE-PEG2000 as the lipid bilayer which have such negative and different zeta potential (-15.1 ~ -28.7 mV) compared to Moses's despite it is known that DSPE-PEG2000 is to screen the charge. It is known that the charge of nanoparticles affect the protein corona significantly. The blood clearance seems have some correlation with the

charge, the more negative surface charge the quicker clearance rate.

Author Response: We thank the reviewer for the critical comments. Nevertheless, we disagree with reviewer on that “*the more negative surface charge the quicker clearance rate*” as suggested by the reviewer, for reasons as follow.

Compared to our PLGA@lipid (zeta-potential of -24.3 mV, blood clearance $t_{1/2}$ of 14.2 h), our 1700kPa@lipid is more negative in zeta-potential (-25.8 mV) (Figure 1e) but exhibited longer circulation lifetime in blood ($t_{1/2}$ of 16.5 h) (Figure 2b-c).

In the work (*Nat. Commun.*, **2018**, 9, 130) by Moses *et al.*, all the nanoparticles (namely, which are nanoliposome (NLP), uncrosslinked nanolipogel (NLG), and crosslinked NLG) exhibited negative zeta-potentials (ranging from -5.9 mV to -9.6 mV, varying relatively by ~ 1.63 times) despite that DOPC (1,2-dioleoyl-sn-glycero-3-phosphocholine), a zwitterionic lipid, is the sole component for their lipid bilayer shell, which again supports our summary (Supplementary Table 6) that, for a core-shell structured nanoparticle, its shell is not the sole factor that determines its zeta-potential and its core matters on this aspect as well.

We agree with the reviewer that our nanoparticles and the nanoparticles in the work (*Nat. Commun.*, **2018**, 9, 130) by Moses *et al.*, though both being core-shell structured and with a lipid bilayer as the shell, differ in zeta-potential range (-15.1 \sim -28.7 mV for the former *versus* from -5.9 \sim -9.6 mV for the latter); nevertheless, this difference in zeta-potential range may arise because of the difference in core (calcium or sodium alginate *versus* acrylamide hydrogel) and that in composition of lipid bilayer shell (DOPC *versus* DOPC:DSPE-PEG₂₀₀₀ = 90:10) between these two nanoparticle systems.

Using polystyrene latex particles that are increasingly negative in surface charge density as model nanoparticles, a previous study (*Eur. J. Pharm. Biopharm.*, **2002**, 54, 165–170) has found that increasing nanoparticle surface charge density increases the total amount of adsorbed proteins but imposes negligible effects on the qualitative and quantitative composition of the adsorbed protein pattern. A similar trend was observed in a review paper (*Chem. Soc. Rev.*, **2012**, 41, 2780–2799) which summarized the influence of nanoparticle surface charge on protein corona for anionic polystyrene nanoparticles (Supplementary Table 7). Back to our DOPC:DSPE-PEG₂₀₀₀ = 90:10 bilayer-coated model nanoparticles, they are unanimously negative in zeta-potential (albeit they differed in specific reading of zeta-potential) (Figure 1e). In fact, 1700kPa@lipid and PLGA@lipid exhibited strikingly different corona compositions (Figure 3f) despite of their closely comparable zeta-potentials (-25.8 mV *versus* -24.3 mV) (Figure 1e). Taken together, these results suggest that the difference in corona composition observed for our model nanoparticles should not be ascribed to their difference in specific reading of zeta-potential. Combined with the fact that our model nanoparticles differ significantly in elasticity, the difference in corona composition observed for them should be attributed to nanoparticle elasticity.

Supplementary Table 7. Qualitative relationships between changes in nanomaterial surface charge and the parameters of the resulting protein corona.

	Parameters of the protein corona			
	Density/ thickness	Identity/ quantity	Conformational change	Affinity
↑ Surface Charge density	Increase	No change	Increase	Increase

Also, the big variance (big error bar) of the amounts of adsorbed proteins (Figure 3a) make the quantitative comparison less meaningful.

Author Response: We thank the reviewer for the critical comments. In this work, the total amount of adsorbed proteins on a nanoparticle (Figure 3a) was determined with a Bradford Protein Assay Kit (Beyotime, China). According to the technical information on Bradford Protein Assay Kit by Thermo Fisher Scientific (<https://www.thermofisher.cn/cn/en/home/life-science/protein-biology/protein-assays-analysis/protein-assays/bradford-assays.html>), one disadvantage of Bradford Protein Assay (also called Coomassie based protein assay) is that Coomassie reagents result in about twice as much protein-to-protein variation as copper chelation-based assay reagents. And this disadvantage should apply to all Bradford Protein Assay Kits by different vendors, rather than being vendor-specific. Therefore, the large variance of amounts of adsorbed proteins in Figure 3a may arise because of the poor sensitivity intrinsic to this assay technique. In future separate work on protein quantification, we would replace Bradford Protein Assay Kit with a technique or techniques capable of providing more accurate results on protein quantification.

The claim "no report to our best knowledge has systematically examined the effects of nanoparticle elasticity on protein corona" is not correct. Tengjisi et al. (Influence of nanoparticle mechanical property on protein corona formation, JCIS, 2022, 606, 1737-1744). They found the protein corona of the stiffest nanocapsules contained the highest amount of complement protein (Complement C3) and immunoglobulin proteins, which contributed to their high macrophage uptake.

Author Response: We thank the reviewer for the critical comments. In revision, we have accordingly revised the sentence “no report to our best knowledge has systematically examined the effects of nanoparticle elasticity on protein corona”. It now reads:

“..... Nevertheless, no report to our best knowledge has systematically examined the effects of nanoparticle elasticity on protein corona, despite that nanoparticle elasticity is crucial in nanoparticle’s physiological fate (both *in vitro*^{8, 9, 11-13} and *in vivo*^{8, 14-19}) and that protein corona formation on nanoparticles of differing elasticity may lead to different changes in nanoparticle size —protein corona formation on hard nanoparticles is manifested as an increase in particle mean diameter^{20, 22, 33, 41} whereas that on

liposomes (known to be elastic and soft) can lead to either an increase²⁶ or reduction^{24, 26} in liposome mean diameter—and can result in different surface coverages of some protein family groups⁴².” (reference 42: *Journal of Colloid and Interface Science*, **2022**, 606, 1737-1744)

Moreover, we have carefully read through this work (*Influence of nanoparticle mechanical property on protein corona formation, JCIS, 2022, 606, 1737-1744*), in which the model nanoparticles are four inorganic silica spherical nanocapsules that are similar both in size (150-180 nm) and in zeta-potential (-2 ~ -4 mV) but differ in elasticity (704 kPa versus 25 MPa versus 459 MPa versus 9.7 GPa). Note that our hardest nanoparticle, PLGA@lipid, has an elasticity of 760 MPa and that our nanoparticles (700kPa@lipid, 1400kPa@lipid, 1700kPa@lipid, and PLGA@lipid) offered a same elasticity range as do these three soft silica nanocapsules. We hence paid extra attention on the corona compositions for those three soft silica nanocapsules. We found that this work revealed significant relative content (>5%) of complement in corona but only on the hard silica nanocapsule (with elasticity of 9.7 GPa); all the three soft silica nanocapsules exhibited <5% relative content (>5%) of complement in corona. Interestingly, significant relative contents (5-20%) of apolipoprotein in corona were observed only on the three soft silica nanocapsules (with elasticity of 704 kPa, 25 MPa, and 459 MPa), which is similar to the significant relative contents (5-20%) of lipoprotein in corona we observed with our four model nanoparticles that offer comparable elasticity range (700kPa@lipid, 1400kPa@lipid, 1700kPa@lipid, and PLGA@lipid) (Figure 3e). Unfortunately, this work (*Influence of nanoparticle mechanical property on protein corona formation, JCIS, 2022, 606, 1737-1744*) did not identify the apolipoproteins on the three soft silica nanocapsules, which prevented us from further comparing the corona apolipoprotein compositions on these three soft silica nanocapsules with those on our aforementioned four nanoparticles.

English needs to be significantly improved. For example, the first two sentences of the abstract, "Nanoparticle elasticity is crucial in nanoparticles' physiological fate. How nanoparticle elasticity does so remains, however, unknown", " a same PEGylated".

Author Response: We thank the reviewer for the critical comments. In revision, we have accordingly checked the language carefully and tried our best to improve the language.

Reviewer #2 (Remarks to the Author):

The authors have sufficiently addressed my comments. I recommend publication of this manuscript.

Author Response: We gratefully thank the reviewer for the positive remarks.